# Calreticulin and integrin alpha dissociation induces anti-inflammatory programming in animal models of inflammatory bowel disease

Masayoshi Ohkuro[1,2,3], Jun-Dal Kim[1,4], Yoshikazu Kuboi[3], Yuki Hayashi[4], Hayase Mizukami[4], Hiroko Kobayashi-Kuramochi[3], Kenzo Muramoto[3], Manabu Shirato[3], Fumiko Michikawa-Tanaka[3], Jun Moriya[3], Teruya Kozaki[3], Kazuma Takase[3], Kenichi Chiba[3], Kishan Lal Agarwala[3], Takayuki Kimura[3], Makoto Kotake[3], Tetsuya Kawahara[3], Naoki Yoneda[3], Shinsuke Hirota[3], Hiroshi Azuma[3], Nobuko Ozasa-Komura[3], Yoshiaki Ohashi[3], Masafumi Muratani [5,6], Keiji Kimura[1,4,7], Ieharu Hishinuma[3] & Akiyoshi Fukamizu[1,4]

Inflammatory bowel disease (IBD), including ulcerative colitis and Crohn's disease, is a chronic intestinal inflammatory condition initiated by integrins-mediated leukocyte adhesion to the activated colonic microvascular endothelium. Calreticulin (CRT), a calcium-binding chaperone, is known as a partner in the activation of integrin α subunits (ITGAs). The relationship between their interaction and the pathogenesis of IBD is largely unknown. Here we show that a small molecule, orally active ER-464195-01, inhibits the CRT binding to ITGAs, which suppresses the adhesiveness of both T cells and neutrophils. Transcriptome analysis on colon samples from dextran sodium sulfate-induced colitis mice reveals that the increased expression of pro-inflammatory genes is downregulated by ER-464195-01. Its prophylactic and therapeutic administration to IBD mouse models ameliorates the severity of their diseases. We propose that leukocytes infiltration via the binding of CRT to ITGAs is necessary for the onset and development of the colitis and the inhibition of this interaction may be a novel therapeutic strategy for the treatment of IBD.

[1] Life Science Center for Survival Dynamics, Tsukuba Advanced Research Alliance, University of Tsukuba, 1-1-1 Tennodai, Tsukuba, Ibaraki 305-8577, Japan. [2] Research Institute, EA Pharma Co., Ltd., 1-1 Suzuki-cho, Kawasaki-ku, Kawasaki-shi, Kanagawa 210-8681, Japan. [3] Tsukuba Research Laboratories, Eisai Co., Ltd., 5-1-3 Tokodai, Tsukuba, Ibaraki 300-2635, Japan. [4] Graduate School of Life and Environmental Sciences, University of Tsukuba, 1-1-1 Tennodai, Tsukuba, Ibaraki 305-8572, Japan. [5] Department of Genome Biology, Graduate School of Comprehensive Human Sciences and Faculty of Medicine, University of Tsukuba, 1-1-1 Tennodai, Tsukuba, Ibaraki 305-8575, Japan. [6] Genome Biology Core, Transborder Medical Research Center, University of Tsukuba, 1-1-1, Tennodai, Tsukuba, Ibaraki 305-8575, Japan. [7] Faculty of Life and Environmental Sciences, University of Tsukuba, 1-1-1 Tennodai, Tsukuba, Ibaraki 305-8572, Japan. These authors contributed equally: Masayoshi Ohkuro, Jun-Dal Kim. Correspondence and requests for materials should be addressed to A.F. (email: akif@tara.tsukuba.ac.jp)

nflammatory bowel disease (IBD) is characterized by chronic recurrent mucosal inflammation of the gastrointestinal tract. The recruitment of leukocytes to areas of inflammation is a crucial process for the pathogenesis of IBD[1,2]. It is well known that the activated integrin α subunits (ITGAs) expressed on the surface of multiple leukocyte types become the key mediators of adhesion and migration via interaction with adhesion molecules such as vascular cell adhesion molecule-1 (VCAM-1) and intercellular adhesion molecule-1 (ICAM-1) on the vascular endothelium[3]. Treatment of IBD patients using monoclonal antibodies raised against the cell surface integrin α 4 (ITGA4) has been shown to be effective[4,5].

In contrast to the cell surface portion, in the intracellular C-terminal region of ITGAs[6], there is a highly conserved amino-acid sequence, KxGFFKR, to which calreticulin (CRT) as a potential integrin regulator binds, resulting in enhanced leukocyte cell adhesion[7–9]. Importantly, using the in situ proximity ligation assay (PLA)[10], we have verified that the interaction of CRT and ITGA4 was specifically observed in leukocytes of the inflamed colonic mucosa and non-inflamed site from the section of patient with ulcerative colitis (UC) (Fig. 1a, b and Supplementary Fig. 1). Here, to clarify whether the inhibition of CRT binding to the cytosolic tail of ITGAs would be effective in anti-inflammatory therapy for IBD patients, we attempted to find a potent chemical

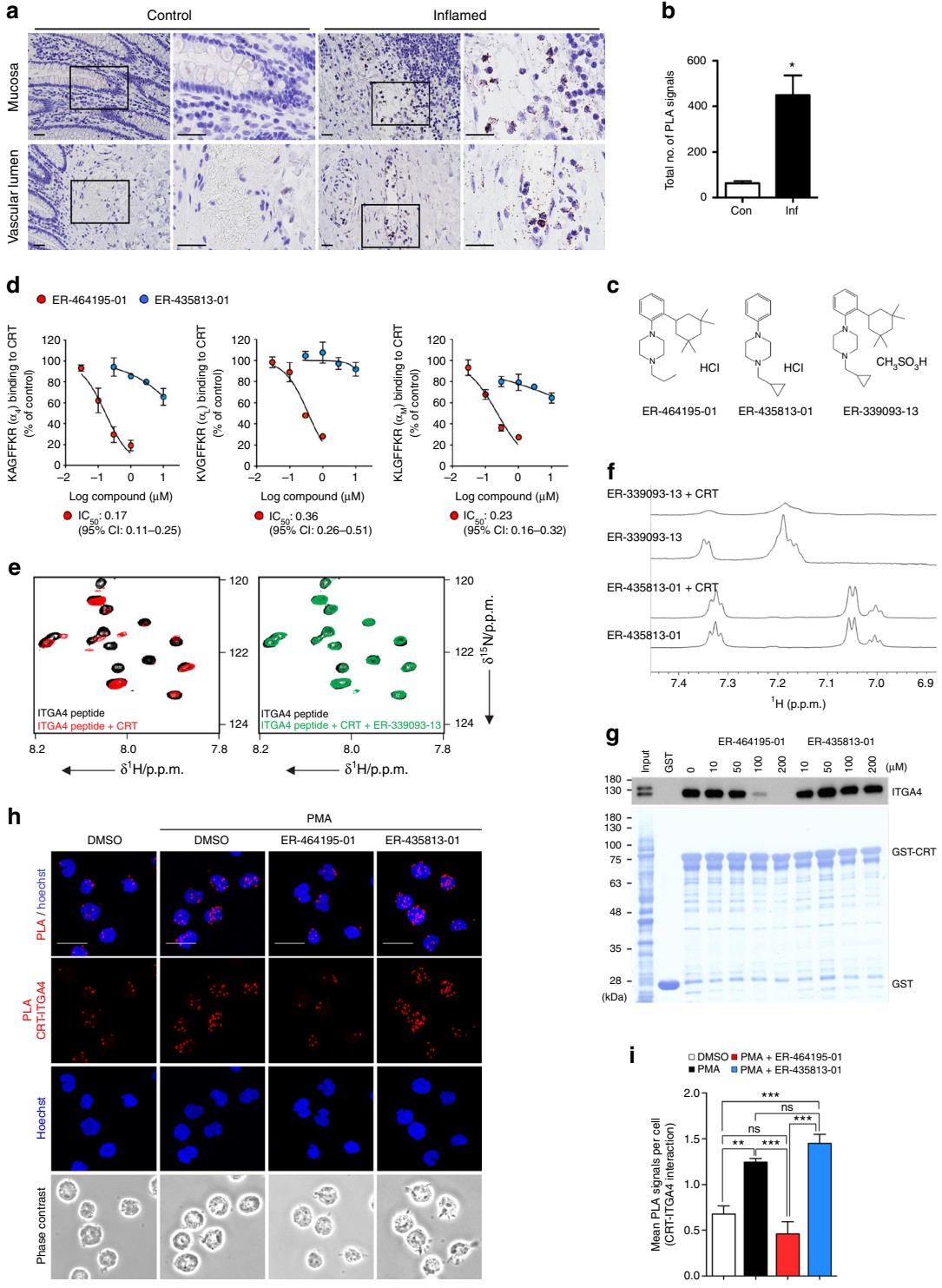

**Fig. 1** ER-464195-01 inhibits interaction of CRT with ITGA. **a** In situ PLA assay for the interaction of CRT with ITGA4 at the mucosa (upper) and the vascular lumen (bottom) in the UC colon. Representative images; non-inflamed control (left) and the inflamed site (right). Right two panels of each site (control or inflamed); enlarged boxes of left two panels in each site. Scale bars, 20 μm. **b** Quantitative analysis of **a**. Results are given as the mean ± SEM of $n = 3$; *$P < 0.05$ (two-tailed $t$ test), the inflamed (Inf) group versus non-inflamed control (Con) group. **c** Chemical structures of small molecules, ER-464195-01, ER-435813-01 and ER-339093-13. **d** $IC_{50}$ values for inhibition of CRT binding to three types of ITGA peptides. Results are given as the mean ± SEM of $n = 3$ independent experiments performed in duplicate. **e** Expansion of two-dimensional $^{15}$N-HSQC spectra of the uniformly $^{15}$N-labeled intracellular domain of ITGA4 (black), with the CRT (red), and with combination of CRT and ER-339093-13 (green). An overall view of the spectra is shown in Supplementary Fig. 4. **f** $^1$H 1D NMR spectra of ER-339093-13 or ER-435813-01 with/without the CRT. Data are representative of at least two independent experiments. **g** In vitro pull-down assay, a dose-dependent-specific inhibition of GST-fused CRT proteins binding to ITGA4 by ER-464195-01 or ER-435813-01. **h** In situ PLA assay for CRT–ITGA4 interaction (red dots) in PMA (100 ng/mL, 30 min)-stimulated Jurkat cells with 5 μM of ER-464195-01 or ER-435813-01. Hoechst 33258 staining (blue); the nucleus. Scale bars, 20 μm. **i** Quantitative analysis of **h**, five independent experiments with at least 500 cells scored in each experiment. Results are given as mean ± SEM of $n = 5$; **$P < 0.01$ and ***$P < 0.0001$ and ns (not significant). Statistical significance was evaluated using one-way ANOVA with Bonferroni's multiple comparison test

compound to inhibit this interaction and functions of leukocyte on the inflammatory conditions in vitro or in vivo, and to assess whether the treatment with compounds exerts protection in mouse models of IBD.

## Results

**Suppression of CRT binding to ITGAs by ER-464195-01**. Using a cell-free binding assay-based high-throughput screening method (Supplementary Fig. 2), we identified the small molecule ER-464195-01 (Fig. 1c) as a potential candidate to inhibit the CRT–ITGAs interaction. As shown in Fig. 1d and Supplementary Table 1, the half-maximal inhibitory concentration ($IC_{50}$) of ER-464195-01 was 0.17, 0.36, and 0.23 μM in the interaction between CRT and the three types of ITGA peptide ($\alpha_4$; KAGFFKR, $\alpha_L$; KVGFFKR and $\alpha_M/\alpha_2/\alpha_5$; KLGFFKR)[8], respectively, in which the first ITGA4 had the lowest $IC_{50}$ value. However, these interactions were not inhibited by a structurally related derivative ER-435813-01 even at concentrations of over 10 μM. The CRT–ITGA4 interaction was also confirmed in surface plasmon resonance analysis performed using intracellular domain of ITGA4 peptide: its dissociation constant of $K_D$ was $2.5 \times 10^{-7}$ (Supplementary Fig. 3). Next, to better understand the inhibitory mechanism of our compounds on the CRT–ITGAs interaction, we performed $^1$H-$^{15}$N-heteronuclear single quantum coherence (HSQC) NMR analysis with $^{15}$N-labeled ITGA4 peptides. The two-dimensional $^{15}$N-HSQC spectrum of ITGA4 peptide showed a different chemical shift with CRT, and returned to spectrum of ITGA4 peptide alone after the pre-treatment with a water-soluble derivative of ER-464195-01, ER-339093-13 (Fig. 1e, Supplementary Figs. 4 and 5). Sequentially, we also acquired a 1D $^1$H NMR spectrum of ER-339093-13 in the presence of CRT. Interestingly, notable line broadening of the ER-339093-13 signal was observed upon the addition of CRT, which had no effect on the signal generated by ER-435813-01 (Fig. 1f), indicating ER-339093-13 associates with CRT.

As an alternative approach for evaluating the inhibitory effects of our compounds for the interaction of CRT with the full-length ITGA, we used the recombinant glutathione S-transferase (GST)-fused CRT protein to pull-down proteins from lysates of the human leukemic T lymphoblast Jurkat cells. The binding of GST–CRT protein to ITGA4 was inhibited by ER-464195-01 in a dose-dependent manner, but not by ER-435813-01 (Fig. 1g). We then used the in situ PLA to investigate whether ER-464195-01 suppresses ITGA4-CRT interaction in activated Jurkat cells. As shown in Fig. 1h, i, PLA signals of their interaction were induced by the phorbol 12-myristate 13-acetate, and were significantly decreased by treatment with 5 μM of ER-464195-01, but not by the same amount of ER-435813-01. Taken together, these data indicated that ER-464195-01 directly binds to CRT, resulting in the blockage of CRT–ITGA4 interaction.

**Inhibitory effects of ER-464195-01 on the leukocyte activity**. During the inflammatory response, the extravasation of leukocytes has been characterized by a series of steps known as adhesion, infiltration and migration, which are initiated by the activation of ITGAs with the stimulation of extra- or intracellular factors[3,11]. To demonstrate the effects of ER-464195-01 on functions of activated leukocytes, we performed the cell adhesion assay in vitro. Divalent cations, such as $Ca^{2+}$ and $Mn^{2+}$, bind to the extracellular domain of ITGAs, and also induce the ITGA-mediated cell adhesion activity[12]. Interestingly, after exogenous activation with $MnCl_2$, the adhesion of T cells to VCAM-1 and neutrophils to ICAM-1 was not inhibited by ER-464195-01 (Fig. 2a,b). By contrast, treating cells with ER-464195-01 dose-dependently inhibited the phorbol 12- myristate 13-acetate (PMA)-induced T cells and the formyl-Methionyl-Leucyl-Phenylalanine (fMLP)-stimulated neutrophils bound to VCAM-1 ($IC_{50} = 0.15$ μM) and ICAM-1 ($IC_{50} = 0.19$ μM), respectively (Fig. 2c, d). Previous studies suggested that CRT is not only localized in the endoplasmic reticulum, but also found on the cell surface[13], where it is associated with the α2-macroglobulin receptor CD91, as a ligand[14]. As shown in Fig. 2e, f and Supplementary Fig. 6, through the flow cytometric analysis of Jurkat cells, we found that the cell surface levels of CRT were significantly reduced by ER-464195-01 alone, but not by ER-435813-01 in the presence of PMA. Our findings revealed that ER-464195-01 targets CRT for the suppressive effects on leukocyte adhesion.

Using the delayed-type hypersensitivity response of oxazolone (OXA)-treated mice[15], we assessed the efficacy of ER-464195-01 in inhibiting the infiltration of inflammatory cells into the colon. Four hours after rectal injection of OXA, the infiltrated lymph cells and glycogen-induced peritoneal exudate cells (PECs) were induced, and were significantly reduced by pre-treatment with oral administration of ER-464195-01 (per os (p.o.), bolus) in a dose-dependent manner (Fig. 2g, h). Taken together, these data supported that ER-464195-01 prevents the adhesion and infiltration of leukocytes into the inflammatory site, which may trigger the inhibition of the onset of IBD.

**Protective effects of ER-464195-01 on DSS-induced colitis**. Next, to examine the protective effects of ER-464195-01 on the onset of IBD, we used the DSS-induced colitis as a model for IBD[15]. In 2% dextran sodium sulfate (DSS)-induced colitis mice, the body weight, disease activity index (DAI) (Supplementary Table 2), and colon length were significantly attenuated by the prophylactic administration of ER-464195-01 at the dose of 10 mg/kg (Fig. 3a–c), suggesting that the onset of IBD can be prevented by ER-464195-01. It has recently been reported that treatment with DSS in mice elevated the intestinal expression of multiple inflammation-related genes[16], the induction patterns of

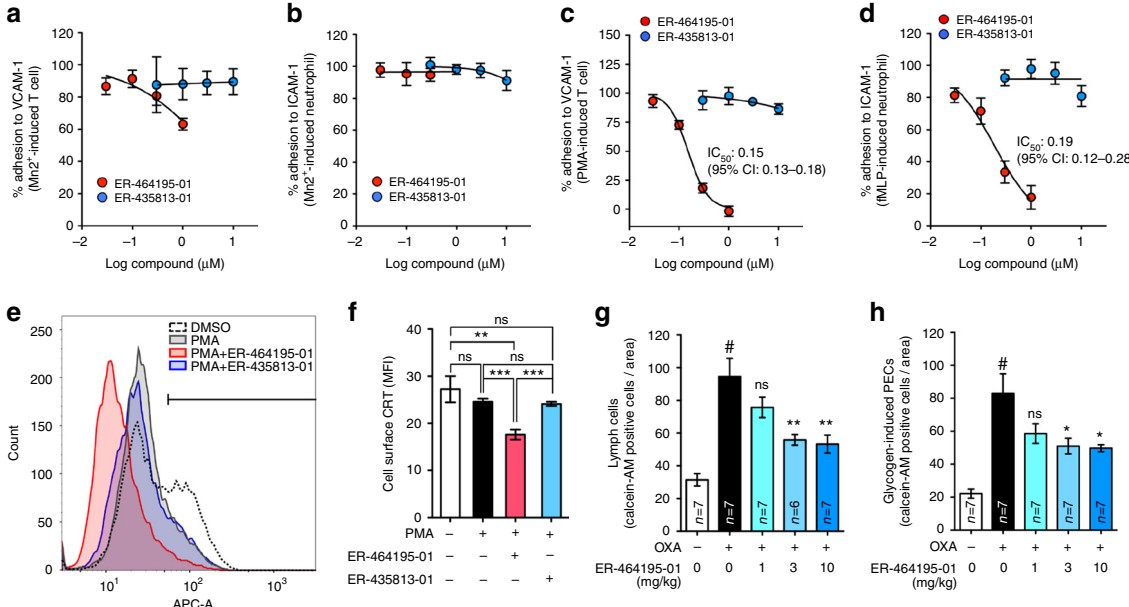

**Fig. 2** ER-464195-01 suppresses the cell adhesion and the migration of leukocytes. **a, b** Inhibition of MnCl$_2$-induced T cells and neutrophils adhere to VCAM-1 **a** and ICAM-1 **b**, respectively, by ER-464195-01 or ER-435813-01. Results are given as the mean ± SEM of $n = 3$ independent experiments performed in duplicate. **c, d** IC$_{50}$ values for inhibition of PMA-induced T cells and fMLP-induced neutrophils adhere to VCAM-1 **c** and ICAM-1 **d**, respectively, by ER-464195-01 or ER-435813-01. Results are given as the mean ± SEM of $n = 3$ independent experiments performed in duplicate. **e**, Flow cytometric analysis for the cell surface CRT in PMA-stimulated Jurkat cells with ER-464195-01 or ER-435813-01. **f**, Quantitative analysis of **e**. Results are given as mean ± SEM of $n = 6$ experiments; *$P < 0.05$, **$P < 0.01$ and ns (not significant). Statistical significance was evaluated using one-way ANOVA with Bonferroni's multiple comparison test. MFI, mean fluorescence intensity. **g, h** In vivo efficacy of ER-464195-01 on OXA-induced lymph cells **g** and glycogen-induced PECs **h** infiltration into OXA-treated colons. Results are given as the mean ± SEM of $n = 6$ or 7; #$P < 0.0001$, OXA group versus control (without the injection of OXA emulsion) (two-tailed $t$ test). *$P < 0.05$, **$P < 0.01$, ***$P < 0.001$ and ns, OXA with ER-464195-01 groups versus OXA group (one-way ANOVA followed by Dunnett's multiple comparison test)

which are highly similar to the colonic transcriptional profiles of UC patients[17,18].

**Transcriptome of DSS-induced colitis with ER-464195-01.** Hence, to gain insight into the molecular mechanisms underlying the prevention of DSS-induced colitis conferred by treatment of ER-464195-01, we performed a comprehensive analysis of ER-464195-01-mediated gene expression changes in DSS-induced colitis using RNA sequencing (RNA-Seq). A total of 81 differentially expressed genes (DEGs) (false discovery rate (FDR) $P < 0.05$, fold change > 2, $n = 6$), including only nine upregulated and 72 downregulated genes, were isolated in intestinal tissues of ER-464195-01 alone compared with the control (normal water) group, suggesting that there was no significant difference between control groups and ER-464195-01 alone (Supplementary Fig. 7 and Supplementary Data 1). We next investigated the relationships among three groups (control, DSS alone, and DSS with ER-464195-01, $n = 5$) by gene expression data clustering with principal component analysis (PCA). As shown in Fig. 3d, PCA plots showed three major distinct clusters, in which transcripts of DSS-induced colitis with ER-464195-01 samples were clearly separated from those of the DSS alone group. To further investigate differentially expressed genes (FDR; $P < 0.05$, fold change > 2) among the three groups, we performed pairwise comparisons of RNA-Seq data using the CLC Genomics Workbench software[19]. As shown in Supplementary Fig. 8, in comparison with the control, 5570 (3483 up- and 2087 downregulated) unique genes were significantly changed in the DSS-induced colitis group. Meanwhile, the data set of DSS-induced colitis with ER-464195-01 groups revealed significant changes in 1808 (837 up- and 971 downregulated) genes as compared with DSS alone. Notably, of 3483 genes upregulated by the DSS treatment, 894 transcripts

were overlapped with 971 genes downregulated by ER-464195-01 (Fig. 3e and Supplementary Data 1).

**Downregulation of inflammation-related gene expression.** To explore whether 894 genes were involved in the inflammatory response of IBD, we then conducted gene ontology (GO) enrichment analysis using the ToppGene Suite[20]. As shown in Fig. 3f, for these genes the molecular function terms revealed the activity of cytokines and chemokines, their receptor binding, the structural constituent of muscle and growth factor activity. Moreover, GO biological processes were mainly enriched for the inflammation-related terms. Notably, the result obtained using the Kyoto Encyclopedia of Genes and Genomes (KEGG) pathway analysis revealed a highly significant association of inflammatory diseases such as cytokine–cytokine receptor interaction, TNF signal pathway, Jak-STAT signaling pathway, rheumatoid arthritis and IBD. Furthermore, we also found significant differences in the expression of key cytokines, chemokines, receptors, inflammatory markers, adhesion factors and intracellular signaling, recently associated with IBD (Fig. 3g)[16,18]. By the above results of GO enrichment analysis, we confirmed the fold changes in the expression of four IBD-related genes that include tumor necrosis factor *Tnfa*, interleukin *Il1b*, *Il6*, and *Il17f*, pro-inflammatory cytokines in colitis and colon cancer[21], by qRT-PCR. As shown in Fig. 3h–k, these genes were significantly induced in DSS-treated colitis group, but not in those treated with ER-464195-01. Moreover, it is known that TNF-α and IL-1β induce activation of the IL-6-STAT3 pathway in immune cells[22,23], where the transcription levels of the plasma acute phase protein serum amyloid A (SAA) family including Saa1, Saa2, and Saa3 were promoted. These proteins trigger a potent inflammatory response in a variety of tissues[24]. Interestingly, the DSS-

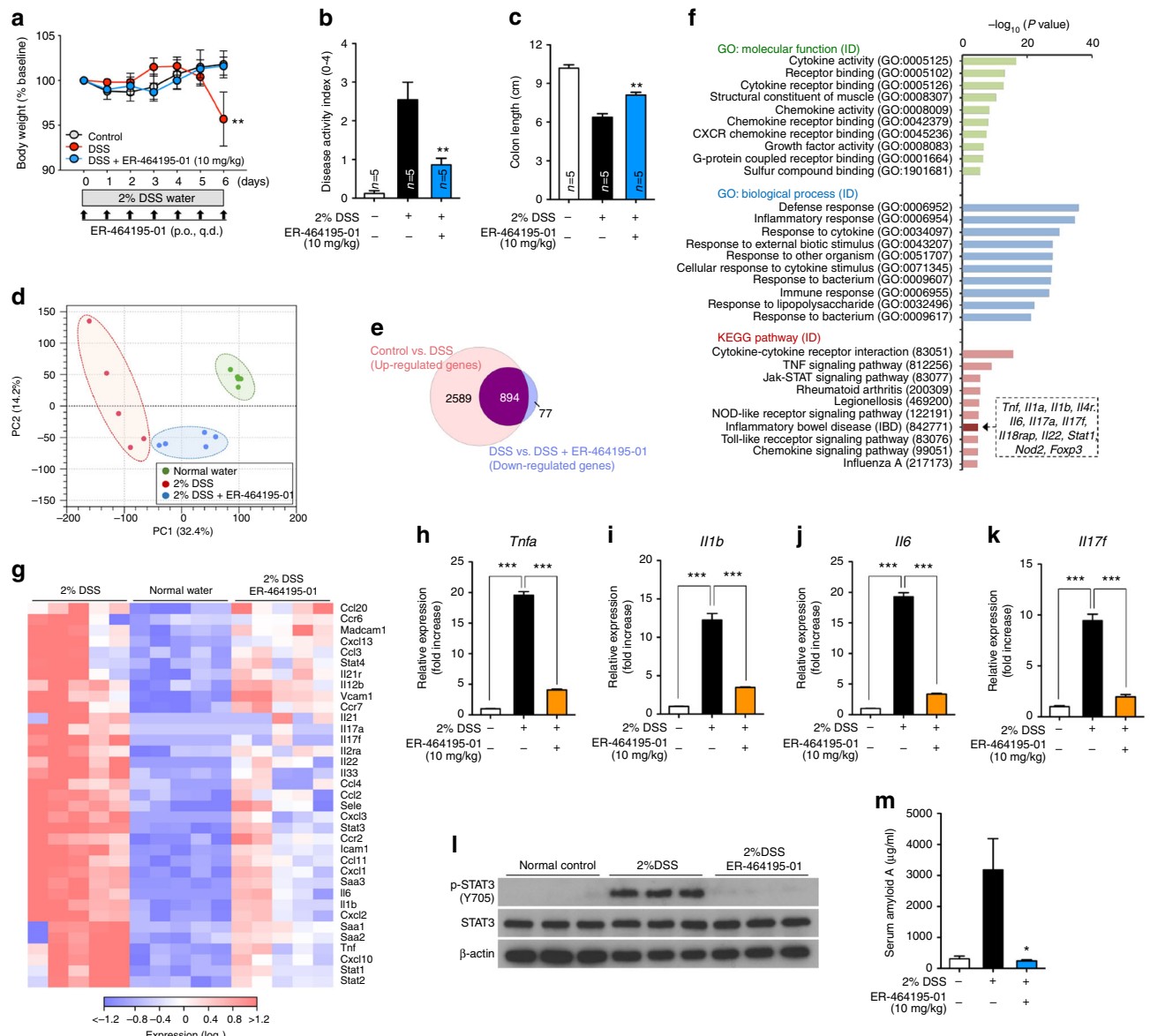

**Fig. 3** ER-464195-01 prevents colitis and downregulates the expression of inflammation-related genes in DSS-treated mice. **a**–**c** Preventing colitis of DSS-treated mice in the prophylactic treatment with ER-464195-01. Body weight **a**, disease activity index (DAI) **b**, and colon length **c**. Results are given as the mean ± SEM of $n = 5$; **$P < 0.01$ versus control group (two-way ANOVA followed by Bonferroni's post test) (**a**), and *$P < 0.05$ and **$P < 0.01$; ER-464195-01-treated groups versus DSS-treated group at day 6 (two-tailed $t$ test) **b**, **c**. **d**–**g**, RNA-seq data. Full gene lists are in Supplementary Data 1. Principal component analysis, PC1 ($x$-axis, 32.4%) and PC2 ($y$-axis, 14.2%) of total variation in DEGs **d**. Venn diagram identifying the intersection (894 genes, magenta) of upregulated gene (DSS alone vs. control group, red) with downregulated genes (DSS with ER-464195-01 vs. DSS alone, blue) (FDR $P < 0.05$, fold change > 2) **e**. Functional enrichment analysis of overlap 894 genes, the negative $\log_{10}$ of the $P$ value. Top 10 enriched GO terms associated with molecular function (green) and biological process (blue), and KEGG pathway analysis (red), list of matched genes with IBD gene sets in the box **f**. Hierarchical cluster analysis of IBD-associated differentially expressed genes (DEGs) among control (normal water), DSS alone and DSS with ER-464195-01 group. Red strip represents high relative expression and blue strip represents low relative expression **g**. **h**–**k**, Quantitative real-time PCR validation for the expression of four genes, *Tnfa* (H), *Il1b* (I), *Il6* (J), and *Il7f* (K). Results are given as the mean ± SEM of $n = 5$; ***$P < 0.0001$ vs. DSS-treated group (two-tailed $t$ test). **l** Western blot for levels of the phosphorylated tyrosine residue of STAT3 in the colon of DSS-treated mice with/without ER-464195-01. Total STAT3 and β-actin were used as loading controls. **m** Concentration of SAA in 2% DSS-induced colitis with/without the prophylactic treatment of ER-464195-01. Results are given as the mean ± SEM of $n = 7$, *$P < 0.05$; ER-464195-01-treated groups versus DSS-treated group at day 6 (two-tailed $t$ test)

induced phosphorylation of STAT3 was successfully inhibited by the treatment with ER-464195-01 (Fig. 3l). As expected, concentrations of SAA were also suppressed by ER-464195-01 in sera of 2% and 3% DSS-induced colitis mice (Fig. 3m and Supplementary Fig. 9). During chronic inflammation, one of the most serious complications of IBD is the development of colorectal cancer[25]. Very interestingly, the expression of tumor-associated

genes including vascular endothelial growth factor *Vegfa*, *Il23a*, *Il23r*, and matrix metalloprotease 9 (*Mmp9*) was almost completely suppressed by the prophylactic administration of ER-464195-01 (Supplementary Fig. 10a–e). Therefore, our data of transcriptome analysis suggested that ER-464195-01 has a role in suppressing the activation of leukocytes and in regulating the expression of inflammation-related genes in IBD model mice.

**Therapeutic effects of ER-464195-01 on murine models of IBD**. Finally, to clarify the therapeutic potential of our compounds as a drug for IBD, we used two kinds of mouse IBD models, CD4 $^+$CD45RB$^{high}$ T-cell transfer colitis[26] and DSS-induced colitis[15]. In T-cell-transferred mice, the therapeutic treatment of ER-464195-01 (p.o., quaque die (q.d.) for 32 days) significantly improved the body

weight loss and stool consistency at the dose of 10 mg/kg (Supplementary Fig. 11). Next, we evaluated the effects of ER-464195-01 on the colitis developed by DSS. After 5 days of exposure to 2% DSS, ER-464195-01-treated mice (p.o., q.d. for 6 days) revealed a significant amelioration of body weight loss, colon shortening, and the DAI score at a dose of 5 mg/kg (Fig. 4a–c). Interestingly,

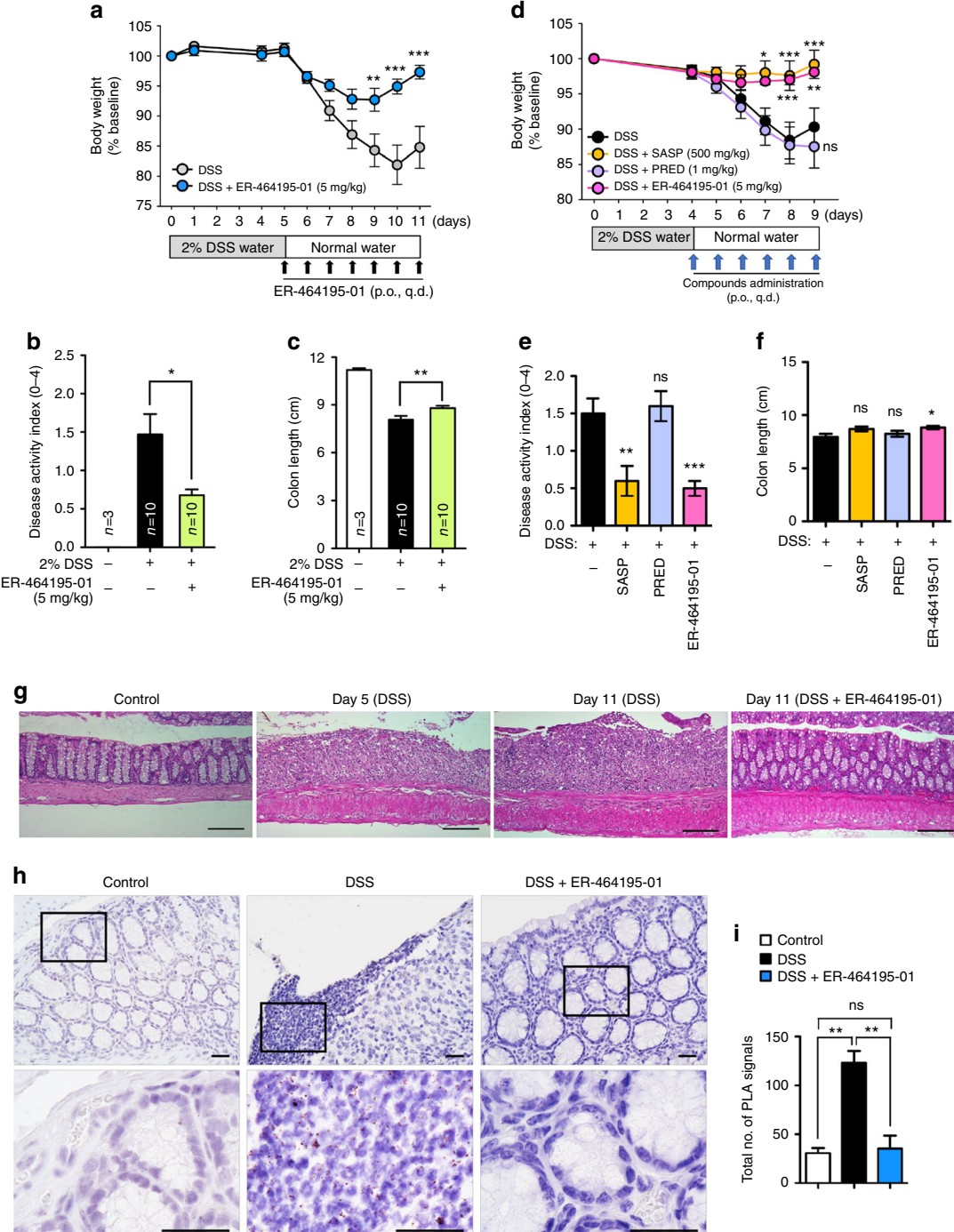

**Fig. 4** ER-464195-01 has therapeutic effects against colitis of DSS-treated mice. **a–c** Improving colitis of DSS-treated mice in the therapeutic treatment with ER-464195-01 ($n = 10$). Body weight **a**, DAI **b**, and colon length **c**. **d–f** Improving colitis of DSS-treated mice in the therapeutic treatment with sulfasalazine (SASP), prednisolone (PRED), or ER-464195-01 ($n = 12$). Results are given as the mean ± SEM of the number of animals; *$P < 0.05$, **$P < 0.01$ and ***$P < 0.001$ vs. DSS alone group (two-way ANOVA followed by Bonferroni's post test) **a, d**, and *$P < 0.05$, **$P < 0.01$, and ***$P < 0.001$; SASP, PRED, or ER-464195-01-treated group vs. DSS-treated group at day 11 **b, c**, or at day 9 **e, f** (two-tailed $t$ test). **g** Hematoxylin and eosin staining for DSS-mediated histological changes. Scale bars, 100 μm. **h** In situ PLA assay for the interaction of CRT with ITGA4 at the mucosa for DSS-mediated histological changes in mice. Bottom panels; enlarged boxes. Scale bars, 20 μm. **i** Quantitative analysis of **h**. Results are given as mean ± SEM of $n = 3$; **$P < 0.01$ and ns (not significant). Statistical significance was evaluated using one-way ANOVA with Bonferroni's multiple comparison test

ER-464195-01 appeared to have an equivalent or higher effect in DSS-induced colitis than oral medications for IBD such as sulfa-salazine at dose of 500 mg/kg (Fig. 4d–f and Supplementary Fig. 12). Moreover, after the DSS treatment, histologic features were regenerated by ER-464195-01 in the time-dependent increase of infiltration of inflammatory cells into the lamina propria and severe colonic damage including the infiltration of mononuclear inflammatory cells, diffuse ulcers and crypt loss (Fig. 4g). To further investigate whether the interaction of CRT and ITGA is induced at leukocytes of the DSS-induced inflamed colonic mucosa, and is inhibited by the therapeutic treatment of ER-464195-01, we examined the levels of their interactions on the large intestine using the PLA assay. Importantly, Fig. 4h, i showed that CRT–ITGA4 interactions were enhanced in the damaged intestine with DSS. By contrast, PLA signals were not detected in the regenerated intestine with ER-464195-01, indicating the inhibition of their interaction. These data showed that ER-464195-01 improves DSS-induced colitis via dissociation of CRT and ITGA4 in leukocytes.

## Discussion

It has been reported that introduction of antibodies or antisense oligonucleotides against CRT into Jurkat cells caused the inhibition of cell adhesion and that embryonic fibroblasts isolated from CRT-deficient mice did not bind to cell adhesion molecules, although the expression of ITGAs was unaltered[7,27]. CRT is known as a ligand for the scavenger receptor CD91 and its expression is downregulated in adenocarcinomas as compared with normal mucosa[28]. On the other hand, we searched levels of gene expression in the RNA-Seq database (GEO accession: GSE83687) of a recent genome-wide association study[29], and confirmed that levels of CRT expression were not significantly different between intestinal tissues of UC or CD patients and controls. Furthermore, the intestinal CRT mRNA and its protein expression were not significantly changed among control, DSS alone, or DSS with ER-464195-01 group in mice (Supplementary Data 1 and Supplementary Fig. 14). Therefore, at least ER-464195-01 is mainly involved in the dissociation of CRT and ITGAs underlying the inflammatory process of IBD.

Over 30% of the patients with IBD including UC and CD will require a surgical intervention during the course of their disease[30], because no single ideal drug therapy is currently available. Thus, there is a serious need for the development of new potential therapeutic agents and treatment strategies for IBD. A recent report suggested that defective signaling of IL-10, an anti-inflammatory cytokine, in a lipopolysaccharide-induced colitis of mice and in macrophages of IBD patients led to dysregulated activation of the NLRP3 inflammasome and production of IL-1β[31]. Intriguingly, despite no evidence concerning the functional contribution of CRT to IL-10 signaling in macrophages, the treatment of ER-464195-01 suppressed expression of Il10 and Nlrp3 (Supplementary Fig. 10f, g and Supplementary Data 1) as well as Il1b (Fig. 3i) in our DSS-induced model. Collectively, our study demonstrates that CRT is a potential therapeutic target for inflammatory disorders via the inhibition of binding to ITGAs by ER-464195-01 in mouse models of IBD. Therefore, it is suggested that the orally active compound provides a novel and promising therapeutic approach for the treatment of IBD.

## Methods

**Antibodies**. The following antibodies were purchased: anti-ITGA4 (EPR1355Y, ab81280), anti-CRT (EPR3924, ab196159), anti-CD4 (3-4F4, ab24894), anti-STAT3 (EPR361, ab109085) from Abcam; anti-CRT (F-4, sc-373863), anti-CRT (N-19, sc-6468) from Santa Cruz Biotechnology; anti-CRT (FMC75, NBP1-97502) from Novusbio; anti-ITGA4 (19676-1AP) from Proteintech; anti-phospho-STAT3 (Tyr705) (D3A7, #9145) from Cell Signaling Technology; anti-b-actin from (M177-3) MBL International Corporation; anti-rabbit IgG-horseradish peroxidase (HRP) (NA934) and anti-mouse IgG-HRP (NA931) from GE Healthcare.

**Compounds**. ER-464195-01 (1-propyl-4-[2-(3,3,5,5-tetramethylcyclohexyl)phenyl] piperazine hydrochloride), ER-435813-01 (1-cyclopropylmethyl-4-phenylpiper-azine hydrochloride), and ER-339093-13 (1-(cyclopropylmethyl)-4-[2-(3,3,5,5-tetramethylcyclohexyl)phenyl] piperazine methanesulfonate) were synthesized (>95% pure) at Eisai Co. Ltd. and are described in patent publication WO2005063705.

**Peptides**. For the high-throughput screen (HTS) cell-free binding assay, we obtained the N-terminally biotinylated ITGA peptides KVGFFKR, KAGFFKR, and KLGFFKR from Operon Biotechnologies K.K. For NMR analysis, intracellular domain of ITGA4, KAGFFKRQYKSILQEENRRDSWSYINSKSNDD, was cloned into a pGEX-2T vector (GE healthcare) using BamHI-EcoRI as a fusion protein tagged with GST. The $^{15}$N-labeled ITGA4 was expressed in *Escherichia coli* (BL21) cultured in $^{15}$N-labeled C. H. L. medium (Chlorella Industry) containing 100 μg/mL ampicillin. In the expression procedure, 1 mM of isopropyl-β-D-thiogalacto-pyranoside (IPTG) and amino-acid solution for the C. H. L. medium were added when the optical density at 600 nm (OD$_{600}$) reached 0.6, and protein expression was induced at 37 °C for 5 h. The cells were harvested, resuspended in 50 mL lysis buffer (20 mM Tris [pH 7.4], 50 mM NaCl, 5 mM EDTA, 1 mM DTT, 0.2% Triton X-100, and one tablet of Complete Tablet$^{TM}$ (Roche)) per 1 L culture, and then disrupted using Bioruptor (COSMO BIO). After centrifugation, the supernatant was mixed with 5 mL (bed volume) Glutathione Sepharose 4B (GE healthcare) at 4 °C for 2 h, and poured into a Glass Econo Column® (Bio-Rad). The flow-through was mixed with 2 mL (bed volume) Glutathione Sepharose 4B (GE healthcare) at 4 °C for 1 h, and poured into the same Glass Econo Column® (Bio-Rad Laboratories, Inc). After washing the resin with 100 mL buffer A (20 mM Tris [pH 7.4], 50 mM NaCl, 5 mM EDTA, 1 mM DTT, and 0.2% Triton X-100), and then 50 mL buffer B (50 mM Tris [pH 7.4], 150 mM NaCl, and 2.5 mM CaCl$_2$), 250 units Thrombin (GE healthcare) was added and incubated for 2.5 h with mixing to separate the GST-tag. After collecting a flow-through fraction containing ITGA4, 16 mL buffer B was added to the column and the eluted fraction containing ITGA4 was collected. The purity of each fraction was confirmed to be more than 90% by sodium dodecyl sulfate-polyacrylamide gel electrophoresis (SDS-PAGE). These fractions were merged and concentrated with Amicon Stirred Cell 50 ml MWCO 1k (Millipore) to 100 μM for NMR-use. For Biacore analysis, the N-terminally biotinylated intracellular domain of the ITGA4 peptide, KAGFFKRQYKSILQEENRRDSW-SYINSKSNDD (>95% pure), was obtained from Operon Biotechnologies K. K.

**Animals**. Balb/c and SCID mice (5- to 6-week-old) were purchased from Charles River Laboratories and CLEA, respectively, in colitis experiments. All mice were housed under a 12 h light–12 h dark cycle, and they had free access to commercial chow and filtered water. Animal care and experimental procedures were performed in the animal facility accredited by the Health Science Center for Accreditation of Laboratory Animal Care and Use of the Japan Health Sciences Foundation. All protocols were approved by the Institutional Animal Care and Use Committee and carried out in accordance with the Animal Experimentation Regulations of Eisai Co., Ltd. and EA Pharma Co., Ltd.

**Cell culture**. Human leukemic T lymphoblast Jurkat cell line was purchased from RIKEN Cell Bank and Dainippon Pharmaceutical Co., Ltd. Cells were tested using a PCR Mycoplasma Detection Set (TAKARA) and were negative for mycoplasma contamination. Cells were cultured and maintained in RPMI 1640 containing 10% heat-inactivated fetal bovine serum (FBS, Cell Culture Technology) and 100 units/mL of penicillin/streptomycin at 37 °C in a humidified 5% CO$_2$ incubator.

**Surface plasmon resonance (SPR) analysis**. Real-time binding and kinetic analysis were performed with Biacore T100 (GE Healthcare) molecular interaction technology through SPR at 25°C. The N-terminally biotinylated intercellular domain of the ITGA4 peptide (KAGFFKRQYKSILQEENRRDSWSYINSKSNDD, >95% pure) was immobilized on a streptavidin-coated sensor chip (SA sensor chip). For immobilization, biotinylated ITGA4 was diluted to 4 μM with HBS-P$^+$ buffer (10 mM 4-(2-hydroxyethyl)-1-piperazineethanesulfonic acid (pH 7.4), 150 mM NaCl, 0.05% surfactant P20) and injected at a flow rate of 10 μL/min. For kinetics analysis of the interaction between CRT (>90% pure) and immobilized ITGA4 peptide, 40 μL of various concentrations of CRT in HBS-P buffer were injected at a flow rate of 20 μL/min. A flow cell not containing peptide was used as blank control. Surface regeneration was achieved via injection of HBS-P$^+$ buffer containing 50 mM NaOH and 500 mM NaCl for 30 s to dissociate bound proteins. Sensorgrams were recorded in real time and analyzed after subtracting the blank control. Data were fitted into the global Langmuir 1:1 binding model using Biacore T100 BIAevaluation 1.1.1 software package (GE Healthcare) in accordance with the manufacturer's instructions.

**HTS cell-free binding assay**. Human CRT was expressed as a maltose binding protein fusion protein in *Escherichia coli* (E. coli, BL21) and was purified with an amylose resin column (New England Biolabs). Each of the 3 biotinylated α integrin peptides (KVGFFKR, KAGFFKR and KLGFFKR) was added to a CRT-coated 96-well plate (Corning) and was then incubated in 10 mM phosphate buffer (PB), pH 7.4, in the presence of compound or vehicle for 1 h at room temperature. After the

addition of HRP–streptavidin (GE Healthcare), the amounts of α integrin peptides bound to CRT were analyzed by measuring the absorbance at 450 nm. Using above HTS cell-free binding assay, Eisai-original HTS library, including one hundred thousand chemical compounds, was screened, resulting in finding of seed compound.

**NMR analyses**. Human CRT protein was cloned into the pET-19b vector (Novagen) using NdeI-XhoI as a His-tag fusion protein. For the PCR reaction, we used the following primers: 5′-GCCTGGCCCATATGGAGCCCGCCGTCTACTT CAAGGAGC-3′ and 5′- GGCATGTACTCGAGCTACAGCTCGTCCTTGGCCT GGC-3′. CRT was expressed in *Escherichia coli* (*E. coli*) BL21 cultured in Luria-Bertani (LB) medium containing 100 μg/mL ampicillin. In the expression procedure, 1 mM IPTG was added when the optical density at 600 nm (OD$_{600}$) reached 0.6, and protein expression was induced at 37 ℃ for 5 h. The cells were harvested, resuspended in 50 ml of lysis buffer (50 mM PB [pH8.0], 200 mM NaCl, 1 mM DTT, and one tablet of Complete Tablet$^{TM}$ (Roche)) per 1 l culture, and then disrupted using Bioruptor (COSMO BIO). After centrifugation, the supernatant was mixed with 3.75 ml (bed volume) Ni-NTA Agarose (Qiagen) at 4 ℃ for 1 h, and poured into a Glass Econo Column® (Bio-Rad). After washing the resin with 37.5 mL buffer C (50 mM PB [pH8.0], 200 mM NaCl, 1 mM DTT, and 5 mM imidazole), 7.5 ml of each buffer C containing 20, 50, 100, 200, 250, and 500 mM imidazole was added into the column in a stepwise manner. CRT was normally eluted in 100, 200, 250, and 500 mM fractions. After merging these fractions, the sample was concentrated with Amicon Ultra MWCO 10000 (Millipore) to 2.5 ml and loaded onto a HiLoad 26/60 Superdex 75 pg column (GE healthcare) equilibrated with buffer D (50 mM PB [pH 8.0], 100 mM NaCl, and 1 mM DTT). The purity of each fraction was confirmed to be more than 90% by SDS-PAGE.

NMR samples were prepared in 200 μL of 20 mM PB, pH 5.5 in 3 mm NMR tube, which contained 10% D$_2$O to provide a field-frequency lock. All NMR experiments were performed on a Bruker Avance 700-MHz spectrometer, equipped with a 5-mm inverse triple resonance cryogenic probe, at 303 K. Two-dimensional $^{15}$N-HSQC spectra for the interaction analysis between $^{15}$N-ITGA4 and CRT/ER-339093-13 were collected with 128 scans and the complex points 128 ($^{15}$N) × 1024 ($^1$H). The carrier position and sweep widths were 117.5 and 33 ppm for the $^{15}$N dimension, and 4.7 and 16 ppm for the $^1$H dimension. Sample concentrations of the experiments were 4 μM $^{15}$N-ITGA4, 4 μM CRT, and 800 μM ER-339093-13. $^1$H 1D spectra for the interaction analysis between CRT and ER-339093-13 or ER-435813-01 were collected with 128 scans and a sweep width of 11261 Hz. Sample concentrations of the experiments were 100 μM ER-339093-13 or ER-435813-01 and 4 μM CRT.

**Plasmid construct and purification of GST fusion proteins**. For the construction of GST–CRT, we used CRT (GenBank accession number NM_004343) sequence to design primer sets with EcoRI/XhoI sites (5′-CCGGAATTCGCCGCCGCATGCT GCTATCCG-3′ and 5′-CCGCTCGAGTGCAGCTCGTCCTTGGCCTG-3′). The cDNA of human CRT was amplified by PCR with total RNA from Jurkat cells as templates. The PCR fragments were digested with EcoRI/XhoI and subcloned into the pGEX6P-2 vector. Transformed *E. coli* BL21 (DE3) cells were incubated with medium containing ampicillin for 3 h at room temperature, and then IPTG (Sigma-Aldrich) was added to a final concentration of 100 μM. After 2 h of induction, harvested cells were sonicated in lysozyme-phosphate-buffered saline (PBS) buffer containing 1 mM phenylmethylsulfonyl fluoride (WAKO) and 1 × protease inhibitor cocktail (Nacalai Tesque). Extracts were incubated with 1% NP-40 at 4 ℃ for 30 min, and were centrifuged at 14,000 rpm at 4 ℃ for 10 min. Recombinant GST-fused CRT was purified on glutathione Sepharose 4B beads (GE Healthcare).

**GST pull-down assay and western blotting**. Jurkat cells were resuspended in lysis buffer (20 mM Tris-HCl (pH 7.4) 150 mM NaCl, 5 mM EDTA, pH 8.0, 1% Nonidet P-40, 5% glycerol, and 1 × protease inhibitor cocktail) for 25 min on ice and centrifuged at 14,000 rpm at 4 ℃ for 10 min. Protein extracts of Jurkat cells were incubated with GST and GST–CRT proteins bound to beads in lysis buffer at 4 ℃ for 4 h, followed by four times for washing with ice-cold PBS buffer and lysis buffer. Bound proteins mixed with 2 × Laemmli sample buffer (Bio-Rad) and heated at 95 ℃ for 5 min and resolved by 10% SDS-PAGE were transferred to an Immobilon-P membrane (Millipore). Membranes were blocked for 45 min at room temperature with 5% skim milk in 1 × TBST (Tris-buffered saline and Tween 20) buffer, and then were incubated with the appropriate primary and secondary antibodies at 4 ℃ overnight and at room temperature for 1 h, respectively. Membranes were incubated with horseradish peroxidase coupled secondary antibody and developed using Luminata Forte Western HRP substrate (Millipore). Immunoreactive bands were detected by exposing to X-ray film (Fujifilm).

**In situ PLA**. To investigate the interaction of CRT–ITGA4 in the colon samples of both UC patients and mice and further in Jurkat cells, in situ PLA was performed using Duolink reagent kit Brightfield and Red (Sigma-Aldrich), respectively. The formalin-fixed tissue section of patient with UC was purchased from The MT Group, Inc. Jurkat cells were fixed with ice-cold methanol for 5 min and then washed using Hanks' balanced salt solution (HBSS) (Thermo Fisher Scientific) for

5 min twice. Methanol-fixed Jurkat cells were incubated with a blocking solution at 37 ℃ for 45 min, then washed with HBSS for 5 min twice. Cells were incubated with the antibody diluent solution including anti-CRT (FMC75, 1:1000 dilution) and anti-ITGA4 (EPR1355Y, 1:1000 dilution) at 4 ℃ overnight. In the usage of colon samples of UC patient and mice, after the deparaffinization and heat-induced antigen retrieval in citrate buffer (pH 6.0) for 15 min, the samples were incubated with a blocking solution inducing 10% donkey serum (Abcam) and 10% Fc Block (Miltenyi Biotec or BD Pharmingen) at 37 ℃ for 60 min, then washed with in situ washing buffer (Sigma-Aldrich) for 5 min twice. The colon samples from UC patient were incubated with a blocking solution including 10% Fc Block (Miltenyi Biotec), anti-CRT (FMC75, 1:25 dilution), and anti-ITGA4 (EPR1355Y, 1:50 dilution) at 4 ℃ overnight. The colon samples from mouse were incubated with a blocking solution including 10% mouse Fc Block (BD Pharmingen), anti-CRT (N-19, 1:40 dilution) and anti-ITGA4 (19676-1-AP, 1:40 dilution) at 4 ℃ overnight. The procedures for the administration of anti-CRT and anti-ITGA4 antibodies, PLA probes (MINUS and PLUS), hybridization, ligation, amplification, and detection were performed according to the manufacturer's protocol. The nuclei were labeled with Hoechst 33,258 (1 μg/mL) for 5 min. Signals of PLA were observed using a fluorescence microscope (BZ-X710, KEYENCE) and a laser-scanning confocal microscope (FluoView FV10i, Olympus). Quantitative analysis of images was performed using Duolink Image Tool software (Sigma-Aldrich).

**Isolation of mouse lymph cells and peritoneal exudate cells**. Preparation of lymph cells; Male Balb/c mice (6-week-old) were sensitized with 150 μL of 3% OXA-ethanol solution placed onto the skin of the abdomen at day 0. At day 3, mice were fasted and at day 4, the lymph nodes that were removed from OXA-sensitized mice were passed through a cell strainer (BD Bioscience) to generate a single-cell suspension that contained lymph cells. Erythrocytes were removed by hypotonic lysis in ammonium chloride buffer (BD Bioscience). Preparation of mouse PECs; 1 mL sample of 10% glycogen (Sigma-Aldrich) in PBS solution was injected into the peritoneal cavity of normal mice. Fourteen hours after the glycogen injection, the infiltrated cells were collected from the peritoneal cavity. Erythrocytes were removed by hypotonic lysis in ammonium chloride (BD Bioscience) buffer to isolated PECs that contained almost neutrophils.

**Cell adhesion assay**. Human neutrophils and T cells were isolated from the peripheral blood of healthy volunteers by means of sedimentation using dextran (Nacalai Tesque) and Ficoll–Paque PLUS (GE Healthcare). Mononuclear cells were collected from the interface between the plasma and the Ficoll layer, and T cells were purified using a nylon fiber column (Wako Pure Chemical Industries). Neutrophils were obtained after hypotonic lysis of contaminating erythrocytes by the treatment with sterile water (within 10 s), as previously reported[9]. Cells were attached to VCAM-1 or ICAM-1 (R&D SYSTEMS)-coated 96-well plate with MnCl$_2$ (2 mM, Sigma-Aldrich), PMA (10 nM, Sigma-Aldrich), or the fMLP (100 nM, Sigma-Aldrich) in the presence of ER-464195-01 or ER-435813-01 for 60 min at 37 ℃. After non-adherent cells were removed by washing, the number of adherent cells was estimated by using the chromogenic substrate 4-nitrophenyl *N*-acetyl-β-D-glucosaminide (Sigma-Aldrich), which is hydrolyzed by the ubiquitous lysosomal enzyme hexosaminidase, and measuring the absorbance at 405 nm[32].

**Flow cytometric analysis of CRT expression on the cell surface**. Jurkat cells were stimulated with PMA (100 ng/mL) and dimethyl sulfoxide or compounds for 30 min. Cells were washed with PBS containing 0.3% FBS and fixed with 3.7% formaldehyde at 37 ℃ for 10 min. After washing with PBS (0.3% FBS), Alexa Fluor 647-conjugated anti-CRT monoclonal antibody and FITC-conjugated anti-CD4 monoclonal antibody were added and incubated at 4 ℃ for 1 h. After washing with PBS (0.3% FBS), the cells were suspended in the same buffer with propidium iodide (1 μg/mL). Expression of CRT and CD4 was measured on the FACSAria II (BD Bioscience) and analyzed by Flowjo software (TreeStar Ashland, OR).

**OXA-induced cell infiltration into the colon of mice**. Male Balb/c mice (6-week-old) were sensitized with 150 μL of 3% OXA-ethanol solution placed onto the skin of the abdomen at day 0. At day 3, all of the mice were fasted and at day 4, isolated lymph cells or PECs were labeled with 2.5 μM calcein-AM (Molecular Probes) for 30 min at 37 ℃, suspended in PBS and intravenously injected into the mice. The OXA was dissolved in peanut oil (Kanto Chemicals) to yield a concentration of 1%. Peanut oil containing 1% OXA was added to an equal volume of ice-cold distilled water and mixed to obtain a 0.5% OXA emulsion. Immediately, 100 μL of OXA emulsion was intrarectally injected into the colon. ER-464195-01 or vehicle (0.5% methyl cellulose solution, 10 mL/kg) was orally administered 30 min before the rectal injection of OXA. Four hours after OXA injection, mice were killed and the colons were removed. The number of calcein-AM-labeled cells that had infiltrated into the colon was estimated by counting the cells placed under a fluorescent microscope (Multi Angle Microscopic System, KEYENCE).

**DSS-induced colitis for prophylactic treatment**. Male Balb/c mice (6–8-week old) were allowed free access to distilled water (Otsuka Pharmaceutical) containing 2 or 3% DSS (molecular weight: 36,000–50,000; MP Biomedicals) for 6 or 7 days to induce colitis. ER-464195-01 or vehicle was orally administered once per day for 6

or 7 days. DAI and colon length were measured on the final day of the DSS treatment. DAI was evaluated as previously described[33,34] with slight modifications (Supplementary Table 2).

**Measurement of SAA in sera**. Serum levels of SAA were measured using an enzyme-linked immunosorbent assay (ELISA) kit (Life Technologies).

**CD4+CD45RBhigh T-cell transfer model of colitis in mice for therapeutic treatment**. CD4+CD45RBhigh T cells were isolated from the splenocytes of female Balb/c mice (6-week-old). The spleens were passed through a cell strainer (BD Bioscience) to generate a single-cell suspension that contained splenocytes. Erythrocytes were removed by hypotonic lysis in ammonium chloride buffer (BD Bioscience). After hemolysis, the CD4+ T cells were separated from the splenocytes by negative selection using a CD4+ T-cell isolation kit (Miltenyi Biotec) and passing the cells over a MACS LS column (Miltenyi Biotec). To isolate CD4+CD45RBhigh T cells, CD4+ T cells were labeled with a phycoerythrin-conjugated mouse anti-CD4 monoclonal antibody (eBioscience) and a FITC-conjugated mouse anti-CD45RB monoclonal antibody (eBioscience). Two-color cell sorting was performed using the FACSAria system (BD Bioscience). The sorted CD4+CD45RBhigh T cells were suspended in PBS and then kept on ice until required for use. At day 0, the CD4+CD45RBhigh T cells ($5 \times 10^5$ cells) were intraperitoneally transferred to female severe combined immunodeficiency (SCID) mice (7-week-old). ER-464195-01 or vehicle was orally administered once per day from day 23 to day 55. Stool consistency and colon wall thickness were determined at day 55. Stool consistency was evaluated by using previously described scoring method[33,34] with slight modifications (Supplementary Fig. 11a).

**DSS-induced colitis for therapeutic treatment**. Female Balb/c mice (9- to 10-week-old) were allowed free access to 2% DSS-containing distilled water from day 0 in order to induce colitis. At day 4 or 5, the 2% DSS-containing distilled water was switched to distilled water, ER-464195-01 or vehicle was orally administered once per day until the final day. DAI and colon length were measured on the final day (day 9 or 11). Colons were removed and fixed in 4% formalin and stained with hematoxylin and eosin and images were obtained using a light microscope, and a digital camera.

**Quantitative real-time PCR analysis**. Total RNA of mouse colon was extracted using ISOGEN (Nippon Gene Co.) and was quantified by NanoDrop (NanoDrop 2000, Thermo Scientific). The first strand complementary DNA was prepared by reverse transcription using ReverTra Ace (Toyobo). qPCR analysis was performed using SYBR Green PCR Master Mix and the Thermal Cycler Dice Real Time System (Takara Bio Inc.). The following conditions were used: 1 cycle for 30 s at 95 °C, and then 40 cycles of 95 °C for 5 s, 60 °C for 30 s, and then 95 °C for 15 s and 60 °C for 30 s, finally 1 cycle for 15 s at 95 °C, and carried out for dissociation analysis. Each gene expression was normalized with *GAPDH* mRNA content and the $\Delta\Delta C_t$ method was used for all real-time PCR analyses. All amplifications were performed in duplicate. Primer sequences are given in Supplementary Table 3.

**Colon lysate preparation and western blotting**. Frozen colons were ground into a fine powder using a Multi-Beads Shocker (Yasui Kikai, Osaka, Japan). They were homogenized in lysis buffer, and then centrifuged at 14,000 rpm at 4 °C for 10 min. Supernatants were mixed with 2 × Laemmli sample buffer and heated at 95 °C for 5 min. Samples were resolved by 10% SDS-PAGE and were transferred to an Immobilon-P membrane (Millipore). Membranes were blocked for 45 min at room temperature with 5% skim milk in 1 × TBST buffer and were incubated with the appropriate primary and secondary antibodies at 4 °C overnight or at room temperature for 1 h. Finally, membranes were incubated with Luminata Forte Western HRP substrate (Millipore), and immunoreactive bands were detected by exposing to X-ray film (Fujifilm).

**RNA preparation and sequencing**. For the RNA sequencing analysis, total RNA was extracted from colon of mice using ISOGEN II reagent (Nippon Gene) according to the manufacturer's instruction. In all, 500 ng of total RNA were ribosomal RNA-depleted using NEBNext rRNA Depletion Kit (New England Biolabs), and converted to Illumina sequencing library using NEBNext Ultra Directional RNA Library Prep Kit (New England Biolabs). The library was validated with Bioanalyzer (Agilent Technologies) to determine size distribution and concentration and sequenced on the NextSeq 500 (Illumina) with paired-end 36-base read option. Reads were mapped on mm10 mouse reference genome and quantified using CLC Genomics Workbench version 9.5.1 and 10.1.1 (QIAGEN). RNA-seq data sets are deposited in the NCBI GEO under accession number GSE98407 and GSE109728.

**Identification of DEGs**. To estimate the expression pattern of transcripts among 2% DSS, 2% DSS with ER-464195-01 and control (normal water) sets, read counts were normalized by calculating number of reads per kilobase per million for each transcript in individual samples using the CLC Genomic Workbench software version 9.5.1 or 10.1.1[35]. Filtering characteristics of fold change −2 to 2 (FDR at P

< 0.05) were used to identify the DEGs. Then, comparison of distinct gene expression patterns was visualized in a clustering heat map and PCA. Finally, a Volcano plots were used to compare gene expression levels in terms of the $\log_2$ fold change.

**Functional enrichment analysis**. To assign GO terms, molecular function and biological process, and KEGG pathway analysis of DEGs among all three groups were performed using ToppGene Suite (http://toppgene.cchmc.org)[20].

**Statistical analysis**. Data are expressed as the means ± S.E.M. All statistical analyses were performed using the Prism5 (GraphPad Software) and statistical significance was determined at the *$P < 0.05$, **$P < 0.01$, and ***$P < 0.0001$ levels using an unpaired t test for comparisons between two groups. For multiple comparisons, we used one-way analysis of variance (ANOVA) followed by a Dunnett's multiple comparison test, a Benferroni's multiple comparison test or two-way ANOVA followed with Bonferroni's post test.

**Data availability**. All data needed to evaluate the conclusions in the paper are present in this study or upon reasonable request from the correspondent authors.

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

## Acknowledgements

We thank N. Yamamoto, M. Watanabe, K. Okano, T. Hida, B. Littlefield, Y. Hasegawa, and M. Ikeda for their encouragement; T. Sakairi, Y. Oda, K. Matsubara, M. Murata, Y. Nishino, and A. Tanaka for dedicating to preclinical pharmacological experiments; K. Nakano, J. Sonoda, A. Inomata, Y. Taketa, and M. Shimada for histological experiments and observations; M. Kaneko, Y. Seki, Y. Yamamura for kind support in in vivo experiments; F. Matsuura and K. Kobayashi for chemical synthesis and analysis of compound; S. Fujimoto, K. Kubara, A. Inoue for useful discussion about structure based drug design; M. Murakami, H. Sasaki, S. Kobayashi, A. Kajiwara, late T. Saeki, Y. Okamoto, O. Asano, H. Hiyoshi, J. Nagakawa, T. Kawano, T. Imai, H. Yamauchi, Y. Kai, T. Iike, and T. Ohbayashi for support and helpful advice throughout the project; H. Nishimura (Department of Life Science, Setsunan University) for helpful technical advice and suggestions; M. Hisada and H. Hayashi for kind cooperation to gain the following funding. This work was supported by the Japan Science and Technology Agency's (JST) Newly extended Technology Transfer Program (NexTEP) (to A.F.) (J13-06). We also thank the members of The Fukamizu laboratory for the helpful discussions.

## Author contributions

M.O., J.-D.K., and A.F. planned and designed the experiments and wrote the manuscript. M.O., J.-D.K, Y.K., Y.H., H.K.-K., K.M., M.S., and T.Ki. designed and conducted the in vitro and in vivo experiments. F.M.-T. performed experiments in the HTS cell-free system. J.M. and T.Ko. designed and performed NMR analysis. K.T., J.M., and K.C. prepared recombinant proteins. K.A. designed and performed the Biacore analysis. M.M., H.M., M.O., and J.-D.K. performed analysis of RNA Sequencing experiment. M.K., T.Ka, N.Y., S.H., H.A., N.O.-K., and Y.O. designed and synthesized ER-339093-13 derivatives. K.K. and I.H. discussed the results and coordinated this work. All authors discussed and approved the manuscript.

## Additional information

**Competing interests:** The authors declare no competing interests.

