## [Peer Review File · Nature Communications]

Reviewers' comments:

Reviewer #1 (Remarks to the Author):

This is an interesting, novel, and well written study, in which the authors investigate the role of Calreticulin (CRT) dissociation from the integrin alpha subunits (ITGAs) in IBD pathogenesis. They demonstrate that a small orally active molecule, named ER-464195-01, inhibits the CRT binding to ITGAs, suppressing the adherence ability of both T cells and neutrophils in vitro.

Moreover, in vivo their transcriptome analysis with the colon of dextran sodium sulfate (DSS)-treated mice reveals that the increased expression of pro-inflammatory genes is down-regulated by ER-464195-01. Strikingly, prophylactic and therapeutic administration of this small molecule to IBD mouse models ameliorated the severity of their diseases, indicating that inhibition of CRT-ITGAs interaction is a novel therapeutic strategy in the treatment of IBD.

The first part of the study is very well done. Biochemical analyses allowed the authors to discover this small molecule ER-464195-01 as a potential candidate to inhibit the CRT-ITGAs interaction. They also demonstrate in vitro that ER-464195-01 selectively targets CRT for the suppressive effects on leukocyte adhesion (T cells and neutrophils).

Nevertheless, in the second part of the study, in which the authors assess whether the treatment with compounds exerts protection in mouse models of IBD, there are major concerns:

1) In the entire manuscript (including introduction and discussion sections) the authors focused their attention on the effects of inhibiting the CRT-ITGA only on leukocyte adhesion and migration through the vasculature during inflammatory conditions. However, it has been reported by Toquet et al in 2007 (Altered Calreticulin expression in human colon cancer: maintenance of Calreticulin expression is associated with mucinous differentiation; *Oncol Rep.* 2007 May; 17(5): 1101-7), that CRT is co-expressed with CD91 also in normal colonic epithelial cells and pericryptic myofibroblasts. These cell types are also known to express several ITGAs. How can the author exclude the effects of ER-464195-01 on other cell compartments other than leukocyte trafficking? Can they perform experiments with irradiated chimera mice, where they inject leukocytes pre-treated in vitro with the compound? Such experiment would exclude the other compartments and at the same time give additional information on the role of CRT in intestinal epithelium.

2) The RNAseq data presented in Figure 3 is nice, however the differentially expressed pathways and genes could just be an epiphenomenon due to the reduced inflammation in the presence of ER-464195-01. In few words, the reduced expression of four IBD-related genes that include tumor necrosis factor *Tnfa*, interleukin *Il1b*, *Il6* and *Il17f*, pro-inflammatory cytokines in ER-464195-01-treated mice may just be the consequence and not the cause of the reduced symptoms of colitis. Beneficial effects of the compound may be exerted by other proteins. For this reasons the authors need to validate some of these genes in vitro and/or in vivo through gain and loss of function studies in the presence of ER-464195-01.

3) It would be important to understand the expression of CRT in UC tissues, and not only CRT interaction with ITGA. Is CRT upregulated in IBD patients? What about Crohn's Disease? Since the authors refers to IBD, it would be good to display PLA also in CD patients. This information may give more clinical relevance to the manuscript. Moreover the authors should report the quantifications of the In situ PLA assays showed in Figure 1A and Suppl. Figure 2, including the sample size (how many patients?), which is completely missing.

4) FACS analysis reported in Figure 2e and 2f is not clearly expressed. The authors claim that the cell surface levels of CRT were significantly decreased under the PMA stimulation, and were further reduced by ER-464195-01. However in the plot and in the relative graph it looks like they just showed the percentage of CRT positive cells, which does not necessarily means that CRT on the cell surface is reduced. It would be important to show the MFI of CRT to better evaluate the expression levels.

5) It is hard to extrapolate from Figure 4h that the in vivo inhibition of CRT-ITGA interaction has been successful upon administration of ER-464195-01. This is an important aspect and cannot be presented at the end of the manuscript as the last figure. The authors must immediately show that

the in vivo inhibition was effective. Figure 4h is almost unreadable and it is hard to see the positivity from those small black points. The in situ PLA is a nice assay, but in my opinion should be accompanied by other assays such as immunofluorescence and colocalization studies, including quantification of co-localization. There are various imaging softwares available (ex. IMARIS) that allow to perform such studies.

Minor comments

1. In Figure 2g and h the authors evaluated the anti-inflammatory effects of ER-464195-01 on the OXA-derived dermatitis. It is not clear why they included this type of analysis on dermatitis, since the entire paper is on IBD. Please, justify and discuss accordingly.
2. The authors should improve the discussion, mentioning the study by Toquen et al (see above for reference).
3. Beside adhesion, also transmigration of leukocyte from the peripheral blood to the inflamed mucosa is an important issue in IBD. The authors should perform also in vitro transmigration of activated human T cells and neutrophils in the presence or absence of their compound to confirm and strengthen their in vivo data. Moreover, it would be interesting to see the effects of ER-464195-01 on leukocytes isolated from the blood of IBD patients.

Reviewer #2 (Remarks to the Author):

The manuscript by Ohkuro et al. from the group of professor Fukamizu is an interesting study about the discovery (by a compound screening) of a patented small molecule inhibitor of the intracellular interaction between calreticulin and the cytoplasmic domain of integrin alpha. This interaction is probed with several methods and the active substance is used, with success, in prophylactic and therapeutic schemes for the treatment of DSS-induced colitis and T cell transfer colitis, being animal models for human inflammatory bowel diseases (IBD). By transcriptome analysis of mouse colon biopsies the upregulation of many inflammatory genes by DSS was found to be significantly attenuated by treatment with the new compound.

In view of about 30 percent non-responders to the current biological treatments of IBD and the increasing "epidemic" of IBD, novel treatments need to be developed and studied. In general, this study fits well within such scheme.

Major criticisms

1. In view of the multiple functions of calreticulin (proofreading effects of misfolded proteins and glyco-biological aspects, binding to the mentioned KLGFFKR motif (also present in nuclear receptors), MHC processing and binding to KxGFFKR in integrins) and the fact that only one aspect of this spectrum was probed, it is difficult to envisage that the presented mechanism is the unique and complete explanation of the in vivo effects in the DSS model. In fact, one would have expected a fourth control group in the RNAseq transcriptome analysis with ER-464195-01 only (e.g. Figure 3g). Such group will provide information about the fact whether maybe other functions of calreticulin are affected and whether and possibly which side-effects of the used compound may be expected.
2. Obviously, to obtain major effects of the compound on various cell types, the used cells need to be strongly activated (PMA is a strong lymphocyte stimulus and fMLP is a strong activator of neutrophils). With such strong stimulants, many intracellular systems, including glycosylation and protein folding, are altered. This implies that the reversal by the compound is not necessarily selectively linked to the assumed interaction. Please delete the word "selectively" (page 6, line 13).
3. The pictures of the in situ PLA-assay are very difficult to assess. I have tried to see the color changes. Whereas, these are clear in the fluorescence pictures in Figure 1g, this is much more

difficult to interpret for the histopathology photographs.

4. In the OXA-ethanol mouse experiment it needs to be clearly stated how the control group was treated. For instance, did the control group receive the complete emulsions without the ER compound? Please specify.

5. Suggestion to change in Figure 1h the indication of the statistics of the histograms. Indeed what is relevant is the significant differences between the PMA-treated groups. The red histogram (PMA stimulus, active inhibitor) is significantly different from PMA stimulus only, whereas the blue histogram is (most probably) not significantly different from the PMA only group. A even better solution would be to indicate on top the pairwise comparisons.

6. In figure 3 panel g (and Supplementary Figure 5), the variability in gene expression levels between individual samples is much larger in the 2% DSS plus compound group versus the two other groups (2% DSS and normal water). As already mentioned above, the group with ER-464195-01 only is missing. However, please also explain what the reason behind the higher variability might be.

7. From supplementary Figure 1, it is clear that the interaction between CRT and ITGA4 is of low affinity. It is relevant to explain whether this has an impact on the high concentrations of the compound needed.

Minor criticisms

1. Page 6, lines 7-8: delete "Recent reports suggested that". The work is from two decades ago.

2. From page 8, line 16 onwards, the numbering of references goes wrong. Reference 22 was forgotten and the following ones are incorrect citations, please correct.

3. Page 9 line 11,to have an.... (replace "the" by "an")

4. Page 11, line 17:Laboratories for the dermatitis experiment.

5. Page 13, line 14: glutathione-Sepharose

6. Page 28, legend third line "c"

Reviewer #3 (Remarks to the Author):

IBD is an unpleasant condition for a significant number of people and current therapies and/or management strategies are deficient. The current manuscript identifies a potential chemical probe that the author claim targets the ITGA(4)/CRT interaction. The submission is broad in scope covering HTS, biophysics, cellular studies, gene expression analyses and animal studies.

Fundamentally, this is a strong paper and considerable efforts have been undertaken to study the mechanism of action of the identified compounds. The outcome is a small molecule tool compound that could certainly be used to learn more about IBD and may also serve as a starting point for further drug discovery efforts. There are however a number of issues that I feel the authors should address before publication can be considered; my expertise is such that my comments centre more towards the biophysical/biochemical studies.

As presented this work could not be reproduced because insufficient details have been given in the materials and methods section e.g. details of plasmid/ expression construct/ sequence and protein expression. Similarly evidence of purity of compounds, peptides and proteins is needed for the biophysical and structural work. No detail on concentrations of reagents (e.g. protein and peptide in NMR)

What was the size and source of the HTS library used.

The binding potency determination is poor – there is no upper or lower asymptote to the "curves" and so the calculation of IC50 is fairly meaningless.

It is not clear why a water soluble derivative of ER-464195-01 (ER-339093-13) was needed for HSQC analyses, if the remaining biophysical and cellular analyses were performed with ER-464195-01, implying ER-464195-01 is also water soluble?

15N-1H HSQC data is unconvincing – the conclusion is that the molecule must target CRT but the shift change on addition of ITGA4 is moderate (with only a few signals changing); the data do not confirm a competitive (orthosteric or allosteric inhibition), although the 1D experiments at least

imply an interaction with CRT.

Quality of image in Fig 1g is insufficient to discern the "red dots"

SPR sensorgram does not saturate with dose so it is hard to see how the Kd could have been determined. It is also not referred to in the main manuscript.

Basic toxicity analyses are not performed e.g. MST, Aimes test or other analyses and the nature of these molecules suggests this would be appropriate e.g. basic nitrogen, aniline group.

Fig 3a why did the authors not test the DSS + compound for longer to ascertain the duration of the protective effect?

It isn't clearly explained why the difference in inhibitory behaviour with Manganese versus PMA/MLP stimulation reveals why the compound selectively targets suppressive effects on adhesion.

Reviewers' comments:

Reviewer #1 (Remarks to the Author):

This is an interesting, novel, and well written study, in which the authors investigate the role of Calreticulin (CRT) dissociation from the integrin alpha subunits (ITGAs) in IBD pathogenesis. They demonstrate that a small orally active molecule, named ER-464195-01, inhibits the CRT binding to ITGAs, suppressing the adherence ability of both T cells and neutrophils in vitro. Moreover, in vivo their transcriptome analysis with the colon of dextran sodium sulfate (DSS)-treated mice reveals that the increased expression of pro-inflammatory genes is down-regulated by ER-464195-01. Strikingly, prophylactic and therapeutic administration of this small molecule to IBD mouse models ameliorated the severity of their diseases, indicating that inhibition of CRT-ITGAs interaction is a novel therapeutic strategy in the treatment of IBD.

The first part of the study is very well done. Biochemical analyses allowed the authors to discover this small molecule ER-464195-01 as a potential candidate to inhibit the CRT-ITGAs interaction. They also demonstrate in vitro that ER-464195-01 selectively targets CRT for the suppressive effects on leukocyte adhesion (T cells and neutrophils).

Nevertheless, in the second part of the study, in which the authors assess whether the treatment with compounds exerts protection in mouse models of IBD, there are major concerns:

- 1) In the entire manuscript (including introduction and discussion sections) the authors focused their attention on the effects of inhibiting the CRT-ITGA only on leukocyte adhesion and migration through the vasculature during inflammatory conditions. However, it has been reported by Toquet et al in 2007 (*Altered Calreticulin expression in human colon cancer: maintenance of Calreticulin expression is associated with mucinous differentiation; Oncol Rep. 2007 May;17(5):1101-7*), that CRT is co-expressed with CD91 also in normal colonic epithelial cells and pericytic myofibroblasts. These cell types are also known to express several ITGAs. How can the author exclude the effects of ER-464195-01 on other cell compartments other than leukocyte trafficking? Can they perform experiments with irradiated chimera mice, where they inject leukocytes pre-treated in vitro with the compound? Such experiment would exclude the other compartments and at the same time give additional information on the role of CRT in intestinal epithelium.
- 2) The RNAseq data presented in Figure 3 is nice, however the differentially expressed pathways

and genes could just be an epiphenomenon due to the reduced inflammation in the presence of ER-464195-01. In few words, the reduced expression of four IBD-related genes that include tumor necrosis factor Tnfa, interleukin Il1b, Il6 and Il17f, pro-inflammatory cytokines in ER-464195-01-treated mice may just be the consequence and not the cause of the reduced symptoms of colitis. Beneficial effects of the compound may be exerted by other proteins. For this reasons the authors need to validate some of these genes in vitro and/or in vitro through gain and loss of function studies in the presence of ER-464195-01.

3) It would be important to understand the expression of CRT in UC tissues, and not only CRT interaction with ITGA. Is CRT upregulated in IBD patients? What about Crohn's Disease? Since the authors refers to IBD, it would be good to display PLA also in CD patients. This information may give more clinical relevance to the manuscript. Moreover the authors should report the quantifications of the In situ PLA assays showed in Figure 1A and Suppl. Figure 2, including the sample size (how many patients?), which is completely missing.

4) FACS analysis reported in Figure 2e and 2f is not clearly expressed. The authors claim that the cell surface levels of CRT were significantly decreased under the PMA stimulation, and were further reduced by ER-464195-01. However in the plot and in the relative graph it looks like they just showed the percentage of CRT positive cells, which does not necessarily means that CRT on the cell surface is reduced. It would be important to show the MFI of CRT to better evaluate the expression levels.

5) It is hard to extrapolate from Figure 4h that the in vivo inhibition of CRT-ITGA interaction has been successfull upon administration of ER-464195-01. This is an important aspect and cannot be presented at the end of the manuscript as the last figure. The authors must immediately show that the in vivo inhibition was effective. Figure 4h is almost unreadable and it is hard to see the positivity from those small black points. The in situ PLA is a nice assay, but in my opinion should be accompanied by other assays such as immunofluorescence and colocalization studies, including quantification of co-localization. There are various imaging softwares available (ex. IMARIS) that allow to perform such studies.

Minor comments

1. In Figure 2g and h the authors evaluated the anti-inflammatory effects of ER-464195-01 on the OXA-derived dermatitis. It is not clear why they included this type of analysis on dermatitis, since the entire paper is on IBD. Please, justify and discuss accordingly.

2. The authors should improve the discussion, mentioning the study by Toquen et al (see above for reference).

3. Beside adhesion, also transmigration of leukocyte from the peripheral blood to the inflamed mucosa is an important issue in IBD. The authors should perform also in vitro transmigration of activated human T cells and neutrophils in the presence or absence of their compound to confirm and strengthen their in vivo data. Moreover, it would be interesting to see the effects of ER-464195-01 on leukocytes isolated from the blood of IBD patients.

Reviewer #2 (Remarks to the Author):

The manuscript by Ohkuro et al. from the group of professor Fukamizu is an interesting study about the discovery (by a compound screening) of a patented small molecule inhibitor of the intracellular interaction between calreticulin and the cytoplasmic domain of integrin alpha. This interaction is probed with several methods and the active substance is used, with success, in prophylactic and therapeutic schemes for the treatment of DSS-induced colitis and T cell transfer colitis, being animal models for human inflammatory bowel diseases (IBD). By transcriptome analysis of mouse colon biopsies the upregulation of many inflammatory genes by DSS was found to be significantly attenuated by treatment with the new compound.

In view of about 30 percent non-responders to the current biological treatments of IBD and the increasing "epidemic" of IBD, novel treatments need to be developed and studied. In general, this study fits well within such scheme.

Major criticisms

1. In view of the multiple functions of calreticulin (proofreading effects of misfolded proteins and glyco-biological aspects, binding to the mentioned KLGFFKR motif (also present in nuclear receptors), MHC processing and binding to KxGFFKR in integrins) and the fact that only one aspect of this spectrum was probed, it is difficult to envisage that the presented mechanism is the unique and complete explanation of the in vivo effects in the DSS model. In fact, one would have expected a fourth control group in the RNAseq transcriptome analysis with ER-464195-01 only (e.g. Figure 3g). Such group will provide information about the fact whether maybe other functions of calreticulin are affected and whether and possibly which side-effects of the used compound

may be expected.

2. Obviously, to obtain major effects of the compound on various cell types, the used cells need to be strongly activated (PMA is a strong lymphocyte stimulus and fMLP is a strong activator of neutrophils). With such strong stimulants, many intracellular systems, including glycosylation and protein folding, are altered. This implies that the reversal by the compound is not necessarily selectively linked to the assumed interaction. Please delete the word “selectively” (page 6, line 13).

3. The pictures of the in situ PLA-assay are very difficult to assess. I have tried to see the color changes. Whereas, these are clear in the fluorescence pictures in Figure 1g, this is much more difficult to interpret for the histopathology photographs.

4. In the OXA-ethanol mouse experiment it needs to be clearly stated how the control group was treated. For instance, did the control group receive the complete emulsions without the ER compound? Please specify.

5. Suggestion to change in Figure 1h the indication of the statistics of the histograms. Indeed what is relevant is the significant differences between the PMA-treated groups. The red histogram (PMA stimulus, active inhibitor) is significantly different from PMA stimulus only, whereas the blue histogram is (most probably) not significantly different from the PMA only group. A even better solution would be to indicate on top the pairwise comparisons.

6. In figure 3 panel g (and Supplementary Figure 5), the variability in gene expression levels between individual samples is much larger in the 2% DSS plus compound group versus the two other groups (2% DSS and normal water). As already mentioned above, the group with ER-464195-01 only is missing. However, please also explain what the reason behind the higher variability might be.

7. From supplementary Figure 1, it is clear that the interaction between CRT and ITGA4 is of low affinity. It is relevant to explain whether this has an impact on the high concentrations of the compound needed.

Minor criticisms

1. Page 6, lines 7-8: delete “Recent reports suggested that”. The work is from two decades ago.

2. From page 8, line 16 onwards, the numbering of references goes wrong. Reference 22 was forgotten and the following ones are incorrect citations, please correct.

3. Page 9 line 11,to have an.... (replace “the” by “an”)

4. Page 11, line 17:Laboratories for the dermatitis experiment.

5. Page 13, line 14: glutathione-Sepharose

6. Page 28, legend third line "c"

Reviewer #3 (Remarks to the Author):

IBD is an unpleasant condition for a significant number of people and current therapies and/or management strategies are deficient. The current manuscript identifies a potential chemical probe that the author claim targets the ITGA(4)/CRT interaction. The submission is broad in scope covering HTS, biophysics, cellular studies, gene expression analyses and animal studies. Fundamentally, this is a strong paper and considerable efforts have been undertaken to study the mechanism of action of the identified compounds. The outcome is a small molecule tool compound that could certainly be used to learn more about IBD and may also serve as a starting point for further drug discovery efforts. There are however a number of issues that I feel the authors should address before publication can be considered; my expertise is such that my comments centre more towards the biophysical/biochemical studies.

As presented this work could not be reproduced because insufficient details have been given in the materials and methods section e.g. details of plasmid/ expression construct/ sequence and protein expression. Similarly evidence of purity of compounds, peptides and proteins is needed for the biophysical and structural work. No detail on concentrations of reagents (e.g. protein and peptide in NMR)

What was the size and source of the HTS library used.

The binding potency determination is poor – there is no upper or lower asymptote to the “curves” and so the calculation of IC₅₀ is fairly meaningless.

It is not clear why a water soluble derivative of ER-464195-01 (ER-339093-13) was needed for HSQC analyses, if the remaining biophysical and cellular analyses were performed with ER-464195-01, implying ER-464195-01 is also water soluble?

15N-1H HSQC data is unconvincing – the conclusion is that the molecule must target CRT but the shift change on addition of ITGA4 is moderate (with only a few signals changing); the data do not confirm a competitive (orthosteric or allosteric inhibition), although the 1D experiments at least imply an interaction with CRT.

Quality of image in Fig 1g is insufficient to discern the “red dots”

SPR sensorgram does not saturate with dose so it is hard to see how the K_d could have been

determined. It is also not referred to in the main manuscript.

Basic toxicity analyses are not performed e.g. MST, Aimes test or other analyses and the nature of these molecules suggests this would be appropriate e.g. basic nitrogen, aniline group.

Fig 3a why did the authors not test the DSS + compound for longer to ascertain the duration of the protective effect?

It isn't clearly explained why the difference in inhibitory behaviour with Manganese versus PMA/MLP stimulation reveals why the compound selectively targets suppressive effects on adhesion.

Response to Reviewers 1:

The reviewer comments are shown in *Times New Roman italics*.

Our responses are shown in Arial font.

“Figure R” indicates figures used in explanation to the reviewers (data not shown in the text).

Reviewer #1 (Remarks to the Author):

This is an interesting, novel, and well written study, in which the authors investigate the role of Calreticulin (CRT) dissociation from the integrin alpha subunits (ITGAs) in IBD pathogenesis. They demonstrate that a small orally active molecule, named ER-464195-01, inhibits the CRT binding to ITGAs, suppressing the adherence ability of both T cells and neutrophils in vitro.

Moreover, in vivo their transcriptome analysis with the colon of dextran sodium sulfate (DSS)-treated mice reveals that the increased expression of pro-inflammatory genes is down-regulated by ER-464195-01. Strikingly, prophylactic and therapeutic administration of this small molecule to IBD mouse models ameliorated the severity of their diseases, indicating that inhibition of CRT-ITGAs interaction is a novel therapeutic strategy in the treatment of IBD.

The first part of the study is very well done. Biochemical analyses allowed the authors to discover this small molecule ER-464195-01 as a potential candidate to inhibit the CRT-ITGAs interaction. They also demonstrate in vitro that ER-464195-01 selectively targets CRT for the suppressive effects on leukocyte adhesion (T cells and neutrophils).

We would like to thank the reviewer for these positive comments about our work. We have revised the manuscript and added new experimentation and discussion. We believe that in its present form, with the addition of this valuable new information, our study will contribute to an improved understanding of the link between the role of CRT-ITGAs interaction and the pathogenesis of IBD. Please find our point-by-point responses to the reviewer’s comments below.

Nevertheless, in the second part of the study, in which the authors assess whether the treatment with compounds exerts protection in mouse models of IBD, there are major concerns:

1) In the entire manuscript (including introduction and discussion sections) the authors focused their attention on the effects of inhibiting the CRT-ITGA only on leukocyte adhesion and migration through the vasculature during inflammatory conditions. However, it has been reported by Toquet et al in 2007 (Altered Calreticulin expression in human colon cancer: maintenance of Calreticulin expression is associated with mucinous differentiation; Oncol Rep. 2007 May;17(5):1101-7), that CRT is co-expressed with CD91 also

in normal colonic epithelial cells and pericryptic myofibroblasts. These cell types are also known to express several ITGAs. How can the author exclude the effects of ER-464195-01 on other cell compartments other than leukocyte trafficking? Can they perform experiments with irradiated chimera mice, where they inject leukocytes pre-treated in vitro with the compound? Such experiment would exclude the other compartments and at the same time give additional information on the role of CRT in intestinal epithelium.

We thank the reviewer for this suggestion. As the reviewer pointed out, it may be possible that ER-464195-01 could affect colonic epithelial cells and pericryptic myofibroblasts, which play roles in inflammatory conditions. To provide additional information on the effect of ER-464195-01 in intestinal epithelium, the idea of injecting irradiated chimera mice with leukocytes pre-treated with the compound is interesting. Unfortunately, this compound is a small molecule with a high diffusion coefficient and has reversible function in leukocytes. Therefore, it is assumed that the inhibitory effect of our compound would be reduced by washing out the leukocytes after pretreatment with the compound, before injecting the cells into irradiated chimera mice. Furthermore, since the binding of the compound to CRT seems to be reversible, even though the compound-pretreated leukocytes would be intravenously injected, compound would immediately diffuse from intracellular region into the vessel and surrounding tissues, and the inhibitory effects would disappear. We would appreciate it if the reviewers could understand our response.

On the other hand, according to previous studies (Ref. 1 and 2), both apheresis therapy and anti- $\alpha 4\beta 7$ integrin antibody were effective in DSS-induced or T cell-transferred colitis mouse models, suggesting the solid possibility that the compound targeting only leukocytes would have the sufficient ameliorative potential in the treatment of colitis. Moreover, to demonstrate the effects of ER-339093-01 (a structurally related derivative of ER-464195-01) on the activity of cell attachment, we performed the cell adhesion assay *in vitro* using three types of cells: T cells, human umbilical vein endothelial cells (HUVECs), and keratinocytes. **[Redacted]**.

[Redacted]

It is known that both CRT and ITGA4 are expressed in HUVECs (Ref. 3 and 4). Furthermore, the adhesion of HUVECs is also induced by the phorbol 12-myristate 13-acetate (PMA) (Ref. 5). We have used an *in situ* PLA assay to investigate whether the binding of CRT to ITGA4 is increased and their interactions are suppressed by ER-464195-01 in activated HUVECs. We were unable to observe PLA signals of their interaction with/without PMA treatment (See Figure R2). Based on the above information and evidence, we believe that ER-464195-01 affects mainly leukocyte trafficking, resulting in the ameliorative efficacy in the colitis models.

Figure R2. *In situ* PLA assay for the interaction of CRT with ITGA4 in the activated HUVECs. (a) Expression of ITGA4 and CRT in HUVECs. Cells were stimulated by PMA (100 ng/ml, 30 min) with DMSO (control), ER-464195-01 (5 μ M) or ER-435813-01 (5 μ M). Activities of ERK1/2 were determined by Western blotting. (b) *In situ* PLA assay for CRT-ITGA4 interaction (red dot) in PMA-stimulated HUVECs with DMSO, 5 μ M of ER-464195-01 or ER-435813-01. Hoechst 33258 staining (blue); the nucleus. Scale bars, 20 μ m.

2) The RNAseq data presented in Figure 3 is nice, however the differentially expressed pathways and genes could just be an epiphenomenon due to the reduced inflammation in the presence of ER-464195-01. In few words, the reduced expression of four IBD-related genes that include tumor necrosis factor *Tnfa*,

interleukin 11b, Il6 and Il17f, pro-inflammatory cytokines in ER-464195-01-treated mice may just be the consequence and not the cause of the reduced symptoms of colitis. Beneficial effects of the compound may be exerted by other proteins. For this reasons the authors need to validate some of these genes in vitro and/or in vitro through gain and loss of function studies in the presence of ER-464195-01.

We thank the reviewer for pointing this out. There is still no conclusive evidence showing that the reduced symptoms of colitis are specifically due to the down-regulation of pro-inflammatory cytokines by ER-464195-01. We also apologize for not being clear enough about the correlation between the compound and the effects on other proteins, to clearly explain the cause of the reduced symptoms of colitis. In contrast, the dysregulated trafficking of leukocytes (neutrophils, lymphocytes, and monocytes) to the intestinal mucosa is required for the initiation and progression of IBD (Ref. 6). It is now recognized that after migrating to the site of inflammation, activated leukocytes produce and release a variety of pro-inflammatory cytokines, such as IL-1, IL-6, IL-8, IL-12, TNF α , and IFN γ , which contribute to the inflammatory reaction (Ref. 7). Our data show that the adherence ability of both T cells and neutrophils was clearly inhibited by ER-464195-01. Therefore, as a consequence, the productions and releases of those pro-inflammatory cytokines may be down-regulated to cause the reduced symptoms of colitis in ER-464195-01-treated mice.

3) It would be important to understand the expression of CRT in UC tissues, and not only CRT interaction with ITGA. Is CRT upregulated in IBD patients? What about Crohn's Disease? Since the authors refers to IBD, it would be good to display PLA also in CD patients. This information may give more clinical relevance to the manuscript. Moreover the authors should report the quantifications of the In situ PLA assays showed in Figure 1A and Suppl. Figure 2, including the sample size (how many patients?), which is completely missing.

We agree with the reviewer. To investigate the difference in CRT expression between intestinal tissues of healthy controls and UC or CD patients, we analyzed the transcription level using the RNA-Seq database (GEO accession: GSE83687, Ref. 8). As shown in Figure R3, a total of 2,811 and 747 differentially expressed genes (DEGs) (false discovery rate (FDR) $p < 0.05$, fold change > 2 , $n = 4$), in which the gene encoding CRT was not included, were isolated in intestinal tissues of UC and CD patients compared with a control group, respectively (See next page, Figure R3). Calculated RPKM (Reads Per Kilobase per Million reads) values for CRT gene were also not significantly different between the control group and UC or CD patients (See next page, Figure R4). Moreover, to more effectively and clearly convey the message of our study, we

added Western blot data to the revised manuscript as Supplementary Figure 14, and described them on page 10. There were no differences in the intestinal CRT expression among control, DSS alone, and DSS with ER-464195-01 groups in the mice.

Figure R3. Gene expression profile of RNA-Seq data distinguishing between intestinal tissues of UC or CD patients and control group. (a, b) Hierarchical clustering of the expression profiles of differentially expressed between control and UC patients (a) or CD patients (b) (adjusted FDR P value < 0.05). (c, d) A volcano plot of differentially expressed transcripts between UC patients and control group (c) and CD patients and control group (d), respectively.

Figure R4. RPKM values of transcripts expressed in control, UC and CD patients. RPKM values were calculated after reads of each sample were mapped onto the respective assembled transcripts. Results are given as the mean \pm SEM of $n = 4$; ns (not significant) vs. control group (two-tailed t -test).

Following the reviewer's suggestion, we investigated the interaction between CRT and ITGA4 in the sections of intestine of a CD patient using the *in situ* PLA assay. Figure R5 shows that PLA signals of the interaction were certainly observed in the inflamed colonic mucosa in the CD patient. However, since a non-inflamed section of the same patient was not obtained, we could not compare inflamed and non-inflamed sites in the same patient. This result indicates that CD would be also a candidate for indication of our compound.

Figure R5. *In situ* PLA assay for the interaction of CRT with ITGA4 at the inflamed colonic mucosa of CD patient. Scale bars, 20 μ m.

As shown in Figure 1a and Supplementary Figure 1a-c of the revised manuscript, *in situ* PLA assays were performed in intestinal sections of three different patients with UC. In accordance with the reviewer's comments, we quantified the average number of PLA signals per three different areas from each patient sample (See revised Figure 1b) in the revised manuscript. Furthermore, we have also added all images that were used for the quantification, as the revised Supplementary Figure 1a-c.

4) FACS analysis reported in Figure 2e and 2f is not clearly expressed. The authors claim that the cell surface levels of CRT were significantly decreased under the PMA stimulation, and were further reduced by ER-464195-01. However in the plot and in the relative graph it looks like they just showed the percentage of CRT positive cells, which does not necessarily means that CRT on the cell surface is reduced. It would be important to show the MFI of CRT to better evaluate the expression levels.

We thank the reviewer for their constructive and kind comments. To better evaluate the CRT levels on the cell surface, we have represented FACS data as the Median Fluorescent Intensity (MFI). As shown in revised Figure 2e and f, the cell surface levels of CRT were not significantly reduced following the PMA stimulation or PMA with a structurally related derivative ER-435813-01, when compared to the DMSO controls. In contrast, the average MIF values of cell surface CRT in PMA with ER-464195-01 were significantly decreased as compared with all conditions. Accordingly, in the revised manuscript, we modified the following sentence: "As shown in Figure 2e, f and Supplementary Figure 6, through the flow cytometric analysis of Jurkat cells, we found the cell surface levels of CRT were significantly reduced by ER-464195-01 alone, but not with ER-435813-01 in the presence of PMA." (See page 6 of the revised manuscript). We have now added a histogram plot (revised Figure 2e) and the quantified MFI values (revised Figure 2f). Moreover, the percentage of CRT-positive cells (dot plot) and the relative graph are shown in the revised Supplementary Figure 6.

5) It is hard to extrapolate from Figure 4h that the *in vivo* inhibition of CRT-ITGA interaction has been successful upon administration of ER-464195-01. This is an important aspect and cannot be presented at the end of the manuscript as the last figure. The authors must immediately show that the *in vivo* inhibition was effective. Figure 4h is almost unreadable and it is hard to see the positivity from those small black points. The *in situ* PLA is a nice assay, but in my opinion should be accompanied by other assays such as immunofluorescence and colocalization studies, including quantification of co-localization. There are various imaging softwares available (ex. IMARIS) that allow to perform such studies.

We agree that this is an important point, and would like to explain the reason why we had chosen the PLA assay to investigate the interaction between CRT and ITGA4. As the reviewer suggested, to investigate the localization of CRT and ITGA4 in Jurkat cells, we have performed immunofluorescent staining experiment with antibodies against CRT and ITGA4 using a confocal microscope. As shown in Figure R6, even though co-localization of CRT and ITGA4 was observed, it has been difficult to find the inhibitory effect of ER-464195-01. This immunofluorescent staining system is very useful for identification of the "co-localization" of CRT with ITGA4, but not

applicable to detect “protein-protein interaction” such as CRT directly binding to ITGA4. Therefore, the phenomenon of increased CRT-ITGA4 interaction by PMA stimulation became detectable using PLA method. Given that the target of ER-464195-01 is the interaction of CRT and ITGAs, we are actually convinced that the compound-mediated dissociation is represented by the change of PLA signals, even without performing other assays.

In the revised version, the data of PLA assay have been improved and showed a higher quality and resolution (See revised Figure 4h and Supplementary Figure 13). In addition, we quantified the average number of PLA signals for three different regions from colon samples: normal controls, DSS alone, and DSS with the therapeutic treatment of ER-464195-01 (See revised Figure 4i). We would like to retain these data at the original position in Figure 4 of the revised manuscript, to conclude the efficacies of ER-464195-01 regarding the body weight, DAI, histology, and further, PLA signal on DSS-induced therapeutic treatment models. We would appreciate it if the reviewers can understand our thought.

Figure R6. Subcellular localization of CRT and ITGA4. Jurkat cells were stimulated with PMA (100 ng/ml) with ER-464195-01 or ER-435813-01 (5 μ M) for 30 min. After staining, co-localization between CRT and ITGA4 was analyzed under a confocal microscope. Scale bars, 10 μ m.

Minor comments

1. In Figure 2g and h the authors evaluated the anti-inflammatory effects of ER-464195-01 on the OXA-derived dermatitis. It is not clear why they included this type of analysis on dermatitis, since the entire paper is on IBD. Please, justify and discuss accordingly.

We would like to apologize for confusing the reviewer. In Figure 2i of the initially submitted manuscript, the purpose was to show the anti-inflammatory efficacies of the compound on not only colitis, but also other indications such as in dermatitis animal models. As the reviewer commented, this study is focused on the anti-inflammatory effects of ER-464195-01 in IBD. Eventually, although we could confirm the potential of ER-464195-01 as anti-dermatitis therapy, we have removed the data on the dermatitis model and concentrated on data regarding IBD in the revised manuscript.

2. The authors should improve the discussion, mentioning the study by Toquen et al (see above for reference).

We thank the reviewer for pointing out this reference. We have added this reference in the revised manuscript as Ref. 29.

3. Beside adhesion, also transmigration of leukocyte from the peripheral blood to the inflamed mucosa is an important issue in IBD. The authors should perform also *in vitro* transmigration of activated human T cells and neutrophils in the presence or absence of their compound to confirm and strengthen their *in vivo* data. Moreover, it would be interesting to see the effects of ER-464195-01 on leukocytes isolated from the blood of IBD patients.

We would like to thank the reviewer for these helpful suggestions. It is well known that the activated leukocytes bind to endothelial cells, which is a very important process as an initial step of the inflammatory response, before leukocytes migrate into the inflamed tissues (Ref. 9 and 10). As shown in Figure 2 g and h, orally administered ER-464195-01 inhibited the invasion activity of both lymph cells and PECs into inflamed colon in *in vivo*. Thereby, the compound is able to block the leukocyte transmigration into the inflamed tissue via the inhibition of the leukocyte–endothelial cell adhesion process. On the other hand, as the reviewer suggested, an experiment using leukocytes isolated from the blood of IBD patients is important. However, unfortunately, it is not so easy to timely obtain the fresh blood samples from patients in the active period, and thus there is a high wall to climb to conduct this experiment. Instead of this approach, we have already confirmed the increase of CRT-IGTA4 interaction in the invading leukocytes at the inflamed colon

tissues of UC patients (See revised Figure 1a and Supplementary Figure 1a-c). Moreover, the Phase II clinical trial is now ongoing to examine the efficacy of the oral administration of the compound in UC patients (<https://clinicaltrials.gov>, ClinicalTrials.gov Identifier: NCT03018054). Here, we would like to confirm the “Proof of concept” of this compound in UC patients in the clinical trials.

References cited:

1. Picarella, D. *et al.* Monoclonal antibodies specific for beta 7 integrin and mucosal addressin cell adhesion molecule-1 (MAdCAM-1) reduce inflammation in the colon of scid mice reconstituted with CD45RB(high) CD4(+) T cells. *J. Immunol.* **158**, 2099-2106 (1997).
2. Tomiyasu, N., Mitsuyama, K., Suzuki, A., Masuda, J., Yamasaki, H., Kuwaki, K., Takaki, K., Kitazaki, S., Sata, M. Development of granulocyte and monocyte adsorptive apheresis in the rat dextran sulfate sodium-induced colitis model. *Methods Find Exp. Clin. Pharmacol.* **29**, 265-268 (2007).
3. Massia, S. P. & Hubbell, J. A. Vascular Endothelial-Cell Adhesion and Spreading Promoted by the Peptide Redv of the liics Region of Plasma Fibronectin Is Mediated by Integrin Alpha-4-Beta-1. *J. Biol. Chem.* **267**, 14019-14026 (1992).
4. Calzada, M. J. *et al.* alpha(4)beta(1) integrin mediates selective endothelial cell responses to thrombospondins 1 and 2 in vitro and modulates angiogenesis in vivo. *Circ. Res.* **94**, 462-470, doi:10.1161/01.Res.0000115555.05668.93 (2004).
5. Wiejak, J., Dunlop, J., Gao, S., Borland, G. & Yarwood, S. J. Extracellular Signal-Regulated Kinase Mitogen-Activated Protein Kinase-Dependent SOCS-3 Gene Induction Requires c-Jun, Signal Transducer and Activator of Transcription 3, and Specificity Protein 3 Transcription Factors. *Mol. Pharmacol.* **81**, 657-668, doi:10.1124/mol.111.076976 (2012).
6. Rivera-Nieves, J., Gorfú, G. & Ley, K. Leukocyte Adhesion Molecules in Animal Models of Inflammatory Bowel Disease. *Inflamm. Bowel. Dis.* **14**, 1715-1735, doi:10.1002/ibd.20501 (2008).
7. Wright, H. L., Moots, R. J., Bucknall, R. C. & Edwards, S. W. Neutrophil function in inflammation and inflammatory diseases. *Rheumatology* **49**, 1618-1631, doi:10.1093/rheumatology/keq045 (2010).
8. Peters, L. A. *et al.* A functional genomics predictive network model identifies regulators of inflammatory bowel disease. *Nat. Genet.* **49**, 1437-1449, doi:10.1038/ng.3947 (2017).

9. Imhof, B. A. & Aurrand-Lions, M. Adhesion mechanisms regulating the migration of monocytes. *Nat. Rev. Immunol.* **4**, 432-444, doi:10.1038/nri1375 (2004).
10. Vestweber, D. How leukocytes cross the vascular endothelium. *Nat. Rev. Immunol.* **15**, 692-704, doi:10.1038/nri3908 (2015).

Response to Reviewers 2:

The reviewer comments are shown in *Times New Roman italics*.

Our responses are shown in Arial font.

“Figure R” indicates figures used in explanation to the reviewers (data not shown in the text).

Reviewer #2 (Remarks to the Author):

The manuscript by Ohkuro et al. from the group of professor Fukamizu is an interesting study about the discovery (by a compound screening) of a patented small molecule inhibitor of the intracellular interaction between calreticulin and the cytoplasmic domain of integrin alpha. This interaction is probed with several methods and the active substance is used, with success, in prophylactic and therapeutic schemes for the treatment of DSS-induced colitis and T cell transfer colitis, being animal models for human inflammatory bowel diseases (IBD). By transcriptome analysis of mouse colon biopsies the upregulation of many inflammatory genes by DSS was found to be significantly attenuated by treatment with the new compound. In view of about 30 percent non-responders to the current biological treatments of IBD and the increasing “epidemic” of IBD, novel treatments need to be developed and studied. In general, this study fits well within such scheme.

We thank the reviewer for these positive comments regarding the general interest and technical soundness of our data. In the revised manuscript, we describe a series of new experiments that we performed following the reviewers’ suggestions; we believe that our conclusions have been strengthened as a result. Moreover, we have added and altered several statements in order to improve the conclusion and the discussion. Below, we provide our point-by-point responses to all of the concerns raised by the reviewer.

Major criticisms

1. In view of the multiple functions of calreticulin (proofreading effects of misfolded proteins and glycobiological aspects, binding to the mentioned KLGFFKR motif (also present in nuclear receptors), MHC processing and binding to KxGFFKR in integrins) and the fact that only one aspect of this spectrum was probed, it is difficult to envisage that the presented mechanism is the unique and complete explanation of the in vivo effects in the DSS model. In fact, one would have expected a fourth control group in the RNAseq transcriptome analysis with ER-464195-01 only (e.g. Figure 3g). Such group will provide information about the fact whether maybe other functions of calreticulin are affected and whether and possibly which side-effects of the used compound may be expected.

We agree that this is an important point. As the reviewer commented, it is known that a multifunctional chaperone protein, calreticulin (CRT), plays several roles including protein folding, maturation, and trafficking in various cell types (Ref. 1 and 2). As already shown in Figure R1 of **Response to Reviewers 1**, a structurally related derivative, ER-339093-01, mainly affects leukocytes adhesion. Furthermore, according to previous studies (Ref. 3 and 4), both apheresis therapy and anti- $\alpha 4\beta 7$ integrin antibody were effective in DSS-induced or T cell-transferred colitis mouse models, suggesting the solid possibility that the compound targeting only leukocytes would have sufficient ameliorative potential in the treatment of colitis. Thus, we believe that the compound effects mainly leukocytes trafficking, resulting in showing the efficacy in the colitis models.

As the reviewer suggested, we also conducted RNA sequencing-based transcriptome analysis in intestinal tissues of vehicle or ER-464195-01 only-treated mice. Interestingly, a total of 81 genes were differentially expressed, including 9 up-regulated and 72 down-regulated genes (false discovery rate (FDR) $p < 0.05$, fold change > 2 , $n = 6$) between the two groups (revised Supplementary Figure 7 and Supplementary Table 3), which were isolated from 49,585 sequenced genes that mapped against the mouse mm10 genome reference. These results indicate that this compound appears to produce only minimal effects on gene transcription in intestinal tissues. Furthermore, the Phase I study (<https://clinicaltrials.gov>, ClinicalTrials.gov Identifier: NCT01221818) to examine safety, tolerability, pharmacokinetics, and pharmacodynamics of oral administration of our compound in healthy subjects has already been completed, and it was found that this compound was well tolerated and safe. Since crucial on-target or off-target side effects due to the compound were not seen, a Phase II study (<https://clinicaltrials.gov>, ClinicalTrials.gov Identifier: NCT03018054) in UC patients is now ongoing.

With respect to the above information, unfortunately we still have not found another function of CRT as the compound target molecule, but unexpected side effects are not observed in the clinical trial of our compound, as described above. The mechanism of action (MOA) and chemical structure of this compound are novel, and therefore we would like to continue to carefully move the clinical trial forward, to confirm both efficacy and tolerability for the patients.

2. Obviously, to obtain major effects of the compound on various cell types, the used cells need to be strongly activated (PMA is a strong lymphocyte stimulus and fMLP is a strong activator of neutrophils). With such strong stimulants, many intracellular systems, including glycosylation and protein folding, are

altered. This implies that the reversal by the compound is not necessarily selectively linked to the assumed interaction. Please delete the word “selectively” (page 6, line 13).

We thank the reviewer for this thoughtful suggestion. We have removed “selectively” from the revised manuscript (page 6, line 15).

3. The pictures of the *in situ* PLA-assay are very difficult to assess. I have tried to see the color changes. Whereas, these are clear in the fluorescence pictures in Figure 1g, this is much more difficult to interpret for the histopathology photographs.

According to the reviewer’s comments, the PLA signals may be more clear in the fluorescent immunohistochemistry (IHC) than in the bright-field IHC staining. However, in the analysis of tissue samples using the fluorescent staining, a common complication is the phenomenon of autofluorescence causing red cells to be observed as false positive when illuminated by the excitation light in the absence of fluorochrome (Ref. 5 and 6). We also have observed the autofluorescence of red cells at the colonic mucosa and vascular lumen of UC patients, in the absent of 1st antibodies in PLA staining (See Figure R1).

Figure R1. Autofluorescence of red cells in the paraffin section of UC patient. White arrow head: red cells. Images were obtained from the fluorescence emitted with excitation (EX) at 470/40 nm and emission (EM) at 525/50 nm (green), EX at 545/25 nm and EM at 605/70 nm (red), and EX at 360/40 nm and EM at 460/50 nm (blue) filters.

In this study, we needed to examine the CRT-ITGA4 interaction in both inflamed tissues and vessels, to confirm the importance of this interaction in leukocytes from the point of the mechanism of action (MOA). We therefore selected not fluorescence but bright-field staining

method. In the revised version, to better show PLA signals from bright-field microscope images, we observed tissue samples from UC patients and DSS-treated mice again using a high resolution lens. We have added very remediated images and quantified the average number of PLA signals (See revised Figures 1a, 1b, 4h and 4i). Moreover, we have provided all images that were used for the quantification in the revised Supplementary Figures 1a-c, and 13.

4. In the OXA-ethanol mouse experiment it needs to be clearly stated how the control group was treated. For instance, did the control group receive the complete emulsions without the ER compound? Please specify.

For our *in vivo* cell infiltration experiments, mice were sensitized with 3% oxazolone (OXA) on day 0 and then they were fasted from day 3. On day 4, lymph cells and PECs were isolated from OXA-sensitized mice, the fluorescence-labeled lymph cells and PECs were injected intravenously to the mice, and then colitis was induced by intrarectal injection of the OXA emulsion. In contrast, control mice were not challenged by the OXA emulsion. Prior to the OXA emulsion injection, mice were pre-treated with/without oral administration of ER-464195-01. We have now included the description of control group in the revised Figure 2g legend.

5. Suggestion to change in Figure 1h the indication of the statistics of the histograms. Indeed what is relevant is the significant differences between the PMA-treated groups. The red histogram (PMA stimulus, active inhibitor) is significantly different from PMA stimulus only, whereas the blue histogram is (most probably) not significantly different from the PMA only group. A even better solution would be to indicate on top the pairwise comparisons.

We thank the reviewer for the kind suggestion. We have added the results of statistical significance above the pairwise comparisons among all groups in the quantitative analysis of *in situ* PLA assay (See the revised Fig. 1i).

6. In figure 3 panel g (and Supplementary Figure 5), the variability in gene expression levels between individual samples is much larger in the 2% DSS plus compound group versus the two other groups (2% DSS and normal water). As already mentioned above, the group with ER-464195-01 only is missing. However, please also explain what the reason behind the higher variability might be.

We appreciate the reviewer's comment. To address the reason behind the high variability in the 2% DSS plus compound, as the reviewer suggested, we analyzed colonic samples in mice that were treated with ER-464195-01 only. In the original manuscript, we revealed that 5,570 and

2,280 unique genes were significantly changed in the DSS-induced colitis group and DSS with ER-464195-01, respectively, as compared with normal controls. However, according to the additional experimental data, only 81 DEGs were identified between ER-464195-01 treatment and normal controls (See Figure R2). These results suggest that at least ER-464195-01 itself does not lead to the increasing of the higher variability of DEGs in the 2% DSS plus ER-464195-01 group versus the two groups (2% DSS and normal water) in our transcriptome analysis.

DEGs	Normal vs ER-464195-01	Normal vs DSS	Normal vs DSS+ER-464195-01	DSS vs DSS+ER-464195-01
Up-regulated genes	9	3,483	1,260	837
Down-regulated genes	72	2,087	1,020	965
Total	81	5,570	2,280	1,802

Differentially expressed genes (DEGs), false discovery rate (FDR) $p < 0.05$, fold change > 2 .

Figure R2. The number of unique DEGs among three groups.

Although the reason remains unknown, our current speculation is as follows: The DSS-induced colitis model is known as a very severe model, which shows rapidly worsening symptoms day by day. As the gene expression sensitively reflects the change of disease severity, there might have been some individuals that showed the insufficient improvement in ER-464195-01-treated 2% DSS group. Therefore, it might be considered that the variability in gene expression levels between individual samples was much larger in the 2% DSS plus compound group than the two other groups (2% DSS and normal water).

7. From supplementary Figure 1, it is clear that the interaction between CRT and ITGA4 is of low affinity. It is relevant to explain whether this has an impact on the high concentrations of the compound needed.

As the reviewer pointed out, the affinity of CRT-ITGA4 interaction was not so high, and the requirement of high concentration of our compound was expected from revised Supplementary Figure 3. We had therefore decided to shift the analysis approach of cell-free system from SPR to the NMR method as shown in revised Figure 1e, Supplementary Figures 4 and 5. Thus, we did not conduct analysis on the efficacy of the compound in regard to interaction of CRT and ITGA4 in SPR experiment.

Minor criticisms

1. Page 6, lines 7-8: delete “Recent reports suggested that”. The work is from two decades ago.

We sincerely regret our inadvertent mistake in writing this sentence. In the revised manuscript, we have changed "Recent reports suggested that" to "**Previous studies** suggested that".

2. From page 8, line 16 onwards, the numbering of references goes wrong. Reference 22 was forgotten and the following ones are incorrect citations, please correct.

We apologize for the incorrect reference numbers, and have corrected them. We thank the reviewer for pointing this out, and we have carefully checked all references in the revised manuscript.

3. Page 9 line 11,to have an.... (replace "the" by "an")

4. Page 11, line 17:Laboratories for the dermatitis experiment.

5. Page 13, line 14: glutathione-Sepharose

6. Page 28, legend third line "c"

We have corrected these errors in reference to the minor comments (3 to 6) in the revised version attached.

References cited:

1. Jiang, Y., Dey, S. & Matsunami, H. Calreticulin: roles in cell-surface protein expression. *Membranes (Basel)* **4**, 630-641, doi:10.3390/membranes4030630 (2014).
2. Krause, K. H. & Michalak, M. Calreticulin. *Cell* **88**, 439-443 (1997).
3. Picarella, D. et al. Monoclonal antibodies specific for beta 7 integrin and mucosal addressin cell adhesion molecule-1 (MAdCAM-1) reduce inflammation in the colon of scid mice reconstituted with CD45RB(high) CD4(+) T cells. *J. Immunol.* **158**, 2099-2106 (1997).
4. Tomiyasu, N., Mitsuyama, K., Suzuki, A., Masuda, J., Yamasaki, H., Kuwaki, K., Takaki, K., Kitazaki, S., Sata, M. Development of granulocyte and monocyte adsorptive apheresis in the rat dextran sulfate sodium-induced colitis model. *Methods Find Exp. Clin. Pharmacol.* **29**, 265-268 (2007).
5. Baschong, W., Suetterlin, R. & Laeng, R. H. Control of autofluorescence of archival formaldehyde-fixed, paraffin-embedded tissue in confocal laser scanning microscopy (CLSM). *J Histochem Cytochem* **49**, 1565-1572, doi:10.1177/002215540104901210 (2001).
6. Prost, S., Kishen, R. E., Kluth, D. C. & Bellamy, C. O. Choice of Illumination System & Fluorophore for Multiplex Immunofluorescence on FFPE Tissue Sections. *PLoS One* **11**, e0162419, doi:10.1371/journal.pone.0162419 (2016).

Response to Reviewers 3:

The reviewer comments are shown in *Times New Roman italics*.

Our responses are shown in Arial font.

“Figure R” indicates figures used in explanation to the reviewers (data not shown in the text).

Reviewer #3 (Remarks to the Author):

IBD is an unpleasant condition for a significant number of people and current therapies and/or management strategies are deficient. The current manuscript identifies a potential chemical probe that the author claim targets the ITGA(4)/CRT interaction. The submission is broad in scope covering HTS, biophysics, cellular studies, gene expression analyses and animal studies. Fundamentally, this is a strong paper and considerable efforts have been undertaken to study the mechanism of action of the identified compounds. The outcome is a small molecule tool compound that could certainly be used to learn more about IBD and may also serve as a starting point for further drug discovery efforts. There are however a number of issues that I feel the authors should address before publication can be considered; my expertise is such that my comments centre more towards the biophysical/biochemical studies.

We thank the reviewer for these encouraging comments, which have significantly helped us to improve our manuscript. We appreciate it to learn that this reviewer considers our science sound. Below we describe how, based on the comments of the reviewers, we improved the presentation of our data, point-by-point.

As presented this work could not be reproduced because insufficient details have been given in the materials and methods section e.g. details of plasmid/ expression construct/ sequence and protein expression. Similarly evidence of purity of compounds, peptides and proteins is needed for the biophysical and structural work. No detail on concentrations of reagents (e.g. protein and peptide in NMR).

We thank the reviewer for raising this point. We have modified the manuscript and added the detailed experimental information in the revised Materials and Methods section (highlighted in blue), which should now be reproducible by other researchers. In this study, we used pure compounds (>95%) and peptides (>95%), which we have also included with their concentrations in the Compound and the Peptide section of the revised Materials and Methods.

What was the size and source of the HTS library used.

For HTS cell-free binding assay, we screened information from the Eisai (Eisai Co., Ltd.,

Japan)-original HTS library, which consisted of one hundred thousand chemical compounds, resulting in our finding the seed compound.

The binding potency determination is poor – there is no upper or lower asymptote to the “curves” and so the calculation of IC50 is fairly meaningless.

We thank the reviewer for this comment. We think that reviewer's comment is correct, and that our calculation of IC50 values is reasonable, for the following reasons: In the revised Figure 1d, results from three independent experiments performed in duplicate are presented. Moreover, IC50 values were calculated with 4 compound concentrations, in which 2 points each (totally 4 points) were plotted before and after the IC50 value. In addition, the 95% confidence interval (CI) values are also properly defined. Calculation of IC50s and CI values was conducted using GraphPad Prism 5 (GraphPad Software Inc., Serial number: GPM5-050316-RAF-4980), which is widely used not only for the calculation of IC50, but also statistical analysis. The reasonable molar ratio of molecules to inhibit protein–protein or protein–peptide interaction is defined by determining the IC50 values, and we also conducted this cell-free experiment to confirm whether our compound could inhibit the interaction CRT and various kinds of integrin peptides in this study. Therefore, we would like to retain and use the original figures and IC50 values as presented in the initially submitted manuscript.

It is not clear why a water soluble derivative of ER-464195-01 (ER-339093-13) was needed for HSQC analyses, if the remaining biophysical and cellular analyses were performed with ER-464195-01, implying ER-464195-01 is also water soluble?

Basically, we conducted both *in vitro* and *in vivo* experiments using ER-464195-01 in this study. ER-464195-01 in DMSO could be diluted in each assay buffer to achieve the desired concentrations that were used in the biophysical and cellular analyses. However, in the HSQC analyses, more than 100 μ M final concentration of compound was needed. The solubility of ER-464195-01 was not high enough in the presence of the HSQC buffer to reach to 100 μ M, and the much more water-soluble ER-339093-13 was suitable and used in the HSQC analysis.

15N-1H HSQC data is unconvincing – the conclusion is that the molecule must target CRT but the shift change on addition of ITGA4 is moderate (with only a few signals changing); the data do not confirm a competitive (orthosteric or allosteric inhibition), although the 1D experiments at least imply an interaction with CRT.

We thank the reviewer for pointing this out. In NMR analysis of this study, although we are not able to conclude whether the inhibition occurs in an orthosteric or an allosteric manner, we believe that 15N-1H HSQC spectra clearly indicate that the inhibition of CRT-ITGA4 interaction by ER-339093-13 is competitive inhibition. In addition, to further clarify the inhibition by the addition of the compound, we have modified the original Figure 1d and presented in revised Supplementary Figure 4, in which the original Figure 1d was separated into two panels, and further the enlarged version were indicated in Figure 1e of the revised manuscript. As shown in the revised Figure 1e and Supplementary Figure 4, the left and the right panels show the interaction between CRT and ITGA4, and the dissociation of its interaction by the compound, respectively.

Quality of image in Fig 1g is insufficient to discern the “red dots”

We greatly appreciate the reviewer’s comment. We have replaced the data in the revised version, in which viewing of the PLA signals (red dots) is improved (See Figure R1).

Figure R1. *In situ* PLA assay for CRT-ITGA4 interaction. PLA signals (red dots) in PMA (100 ng/ml, 30 min)-stimulated Jurkat cells with 5 μ M of ER-464195-01 or ER-435813-01. Hoechst 33258 staining (blue); the nucleus. Scale bars, 10 or 20 μ m.

SPR sensorgram does not saturate with dose so it is hard to see how the K_d could have been determined. It is also not referred to in the main manuscript.

To study CRT binding to ITGA4, the biotinylated ITGA4 was immobilized on the SA sensor chip surface, and solutions of CRT were injected at various concentrations. Resulting sensorgrams (revised Supplementary Figure 3) showed rapid, concentration-dependent increases in Resonance Units (RU) reflective of CRT binding (association), followed by

concentration-dependent decreases in RU during the CRT washout phase, reflecting loss of bound CRT (dissociation). Since the concentration of CRT stock solution was 1.5 mg/mL (30 μ M) in water, maximum concentration of injected CRT was limited to 2 μ M (i.e. 15-fold dilution) to obtain the precise sensorgrams without artificial effects. For the above reason, we could not add high enough concentration of CRT to reach the amount required by the equation.

With regard to the calculation method of dissociation constant K_D value, there are two methods for obtaining K_D value, in Biacore T100 BIAevaluation 1.1.1 software package, “Kinetics analysis” and “Affinity analysis” according to the manufacturer’s instructions. When using “Kinetics analysis” the sensorgrams equation is not needed, and the dissociation constant K_D value can be calculated by dividing the association (k_a) by dissociation (k_d) rate constant, in the following the global Langmuir 1:1 binding model. An appropriate equation to fit the experimental results was chosen based on the best fit for CRT binding to the ITGA4 from the global Langmuir 1:1 binding model, as follows:

$$dR/dt = k_a \cdot \text{Conc} \cdot (R_{\text{max}} - R) - k_d \cdot R \quad (\text{Equation, Ref.1})$$

Where dR/dt is the rate of formation of surface associated complexes, i.e., the derivatives of the observed response curve, C is the constant concentration of ligate in solution, R_{max} is the capacity of the immobilized ligand surface expressed in resonance units, and $(R_{\text{max}} - R)$ is equivalent to the number of unoccupied surface binding sites. By fitting the data to the global Langmuir 1:1 binding model with X^2 , T-values, and U values of the fit being <15, >5, and <15, respectively, the following rate constant values were estimated from the experimental curves: k_a , $1.54 \times 10^4 \text{ M}^{-1}\text{s}^{-1}$ (association rate constant) and k_d , $3.88 \times 10^{-3} \text{ s}^{-1}$ (dissociation rate constant). Rate constants were then used to calculate the K_D (dissociation constant, k_d/k_a) as $2.5 \times 10^{-7} \text{ M}$. Using this approach, K_D can be determined using non-equilibrium binding sensorgrams in accordance with the manufacturer’s instructions.

Quality of data sets and reliability of global fitting in SPR analyses were verified automatically by the BIAevaluation program version 1.1.1, which has three embedded tools for assessing significance of reported constants. First, closeness of fit is established by minimized deviation of small residual segments of experimental data from idealized fitting, indicating that residuals scatter randomly around zero over ranges that correspond to short-term noise in the detection system. Closeness of fit is characterized by X^2 values; in our analyses, only fits with $X^2 < 15$ were accepted. U values represent a second indicator of parameter significance representing validity of calculated rate constants; U values are determined by testing fitting dependence on correlated variations between selected variables. Lower U values indicate greater confidence that reported

kinetic constants contain useful information; in our analyses, only U values <15 were accepted (Ref. 2 and 3). Finally, T-values are estimates of the sensitivity of fitting to changes in the associated parameter. High T-values correspond to low standard errors; in our analyses, only T-values >5 were accepted.

In the revised version of the manuscript, we carefully modified the Method of “Surface plasmon resonance (SPR) analysis” and Figure legend of the revised Supplementary Figure 3.

Basic toxicity analyses are not performed e.g. MST, Aimes test or other analyses and the nature of these molecules suggests this would be appropriate e.g. basic nitrogen, aniline group.

We thank the reviewer for pointing this out. As explained above (see **Response to Reviewers 1 and 2**), we have already completed the Phase I study (<https://clinicaltrials.gov>, ClinicalTrials.gov Identifier: NCT01221818) in healthy subjects, in which it was shown that the compound was well tolerated and safe. Since crucial on-target or off-target side effects due to the compound were not seen, the Phase II study (<https://clinicaltrials.gov>, ClinicalTrials.gov Identifier: NCT03018054) in UC patients is now ongoing. As the mechanism of action (MOA) and chemical structure of this compound are novel, we would like to continue to carefully move the clinical trial forward, to confirm both efficacy and tolerability.

Fig 3a why did the authors not test the DSS + compound for longer to ascertain the duration of the protective effect?

As shown in the revised Supplementary Figure 9, ER-464195-01 was also effective in long-term experiment, in which the duration extended until day 7 to show severe DAI with >15 % body weight loss. In Figure 3, to identify the gene expression change by the prophylactic treatment of ER-464195-01 on the onset of colitis, we decided to end the experiment on day 6, when the significant body weight loss was observed in DSS control group.

It isn't clearly explained why the difference in inhibitory behaviour with Manganese versus PMA/MLP stimulation reveals why the compound selectively targets suppressive effects on adhesion.

Although Induction of leukocyte adhesiveness is observed in a similar fashion by addition of either fMLP, PMA, or Manganese stimulus, each stimulating cascade is different. In short, leukocytes activators such as fMLP and protein kinase C activator PMA induce the leukocyte adhesiveness via upregulation of intracellular signal, while the integrin activator Manganese binds to the extracellular domain of integrin and forces the integrins into an active conformation to

expose the active epitope (Ref. 4 and 5). In the intracellular C-terminal region of ITGAs, there is a highly conserved amino acid sequence, KxGFFKR, to which CRT as a potential integrin regulator binds, resulting in enhanced leukocyte cell adhesion in the presence of PMA stimulation (Ref. 6). Thereby, ER-464195-01 is not able to suppress Manganese-stimulated leukocytes adhesion.

References cited:

1. Oshannessy, D. J., Brighamburke, M., Soneson, K. K., Hensley, P. & Brooks, I. Determination of Rate and Equilibrium Binding Constants for Macromolecular Interactions Using Surface-Plasmon Resonance - Use of Nonlinear Least-Squares Analysis-Methods. *Anal. Biochem.* **212**, 457-468, doi:DOI 10.1006/abio.1993.1355 (1993).
2. Canziani, G. A., Klakamp, S. & Myszka, D. G. Kinetic screening of antibodies from crude hybridoma samples using Biacore. *Anal. Biochem.* **325**, 301-307, doi:10.1016/j.ab.2003.11.004 (2004).
3. Safsten, P., Klakamp, S. L., Drake, A. W., Karlsson, R. & Myszka, D. G. Screening antibody-antigen interactions in parallel using Biacore A100. *Anal. Biochem.* **353**, 181-190, doi:10.1016/j.ab.2006.01.041 (2006).
4. Chen, J. F., Salas, A. & Springer, T. A. Bistable regulation of integrin adhesiveness by a bipolar metal ion cluster. *Nat. Struct. Biol.* **10**, 995-1001, doi:10.1038/nsb1011 (2003).
5. Schurpf, T. & Springer, T. A. Regulation of integrin affinity on cell surfaces. *EMBO J.* **30**, 4712-4727, doi:10.1038/emboj.2011.333 (2011).
6. Coppelino, M. G. *et al.* Calreticulin is essential for integrin-mediated calcium signalling and cell adhesion. *Nature* **386**, 843-847, doi:10.1038/386843a0 (1997).

REVIEWERS' COMMENTS:

Reviewer #2 (Remarks to the Author):

The authors have done an excellent job with the review of their manuscript and answered in a scientifically sound, reserved and kind way to all comments.

This is how real academic work brings research forward.

Congratulations

Reviewer #4 (Remarks to the Author):

In this paper the authors study the effects of a small molecule inhibitor of the intracellular interaction between calreticulin and the cytoplasmic domain of integrin alpha. This interaction is tested with several methods and the active compound is used, in prophylactic and therapeutic protocols for the treatment of DSS-induced colitis and T cell transfer colitis). They also show, by transcriptome analysis of mouse colon biopsies that the upregulation of many inflammatory genes by DSS was significantly inhibited by treatment with this compound.

Since only 30% of IBD patients show a response to biological therapy one year after initiation of treatment and the increasing incidence of IBD, novel treatments are needed. Therefore, this study is novel and highly significant using a novel small molecule compound that targets a very relevant pathway.

This is a resubmission and the authors have done an outstanding job in responding to the excellent criticisms of the original reviewers, modify the paper accordingly, and added new experiments as requested. The paper is therefore, significantly improved.

I will focus my comments on the major issues raised by reviewer 1 dealing with the in vivo experiments.

1) The authors have satisfactorily replied to this criticism; although it would be important to know whether the compound acts on other cells beside leukocytes and endothelial cells, I do not believe that this is a critical issue. For example, at the present time we still don't know the precise mechanism of action of anti-TNF therapies nor their precise cellular target(s). Nevertheless, these drugs represent the gold standard for biological therapy of IBD patients.

2) I am satisfied with the response to this issue. It is fine to show the ability of this compound to suppress proinflammatory cytokine genes. To demonstrate which protein specifically mediate the anti-inflammatory effects of this compound is beyond the scope of this report and not necessarily a critical issue.

3) and 4) In my opinion the authors have satisfactorily responded to these issue and provided additional data.

5) The response here is appropriate. I believe that the authors should decide the order in which the data and figures are presented.

Since the authors employ mouse models of colitis that have limited relevance to human IBD and the results in patients are not compelling, the title of the paper should be modified to: Dissociation of calreticulin and integrin alpha induces anti-inflammatory programming in experimental murine colitis. They should also lessen the reference to human IBD through the Discussion.

Reviewer #5 (Remarks to the Author):

Overall this is a very nice work and I highly recommend acceptance as it is now in the revised form.

I only checked the arguments and the rebuttal of reviewer 3 (biophysical (small molecule issue)). I think the authors have addressed all questions nicely and improved the clarity of the presentation.

No further changes are necessary.

Response to Reviewers:

The reviewer comments are shown in *Times New Roman italics*.

Our responses are shown in Arial font.

REVIEWERS' COMMENTS:

Reviewer #2 (Remarks to the Author):

The authors have done an excellent job with the review of their manuscript and answered in a scientifically sound, reserved and kind way to all comments.

This is how real academic work brings research forward.

Congratulations

We would like to thank the reviewer for these encouraging comments. It is our pleasure if this study is helpful for researches and related communities of inflammatory bowel disease.

Reviewer #4 (Remarks to the Author):

In this paper the authors study the effects of a small molecule inhibitor of the intracellular interaction between calreticulin and the cytoplasmic domain of integrin alpha. This interaction is tested with several methods and the active compound is used, in prophylactic and therapeutic protocols for the treatment of DSS-induced colitis and T cell transfer colitis). They also show, by transcriptome analysis of mouse colon biopsies that the upregulation of many inflammatory genes by DSS was significantly inhibited by treatment with this compound.

Since only 30% of IBD patients show a response to biological therapy one year after initiation of treatment and the increasing incidence of IBD, novel treatments are needed. Therefore, this study is novel and highly significant using a novel small molecule compound that targets a very relevant pathway.

This is a resubmission and the authors have done an outstanding job in responding to the excellent criticisms of the original reviewers, modify the paper accordingly, and added new

experiments as requested. The paper is therefore, significantly improved.

I will focus my comments on the major issues raised by reviewer 1 dealing with the in vivo experiments.

1) The authors have satisfactorily replied to this criticism; although it would be important to know whether the compound acts on other cells beside leukocytes and endothelial cells, I do not believe that this is a critical issue. For example. at the present time we still don't know the precise mechanism of action of anti-TNF therapies nor their precise cellular target(s). Nevertheless, these drugs represent the gold standard for biological therapy of IBD patients.

2) I am satisfied with the response to this issue. It is fine to show the ability of this compound to suppress proinflammatory cytokine genes. To demonstrate which protein specifically mediate the anti-inflammatory effects of this compound is beyond the scope of this report and not necessarily a critical issue.

3) and 4) In my opinion the authors have satisfactorily responded to these issue and provided additional data.

5) The response here is appropriate. I believe that the authors should decide the order in which the data and figures are presented.

Since the authors employ mouse models of colitis that have limited relevance to human IBD and the results in patients are not compelling, the title of the paper should be modified to: Dissociation of calreticulin and integrin alpha induces anti-inflammatory programming in experimental murine colitis. They should also lessen the reference to human IBD through the Discussion.

We greatly appreciate the reviewer's comments, which are significantly helpful for us to improve our manuscript. We have carefully considered and replaced the title as follow: Dissociation of calreticulin and integrin alpha induces anti-inflammatory programming in animal models of inflammatory bowel disease.

In addition, as the reviewer suggested, we have altered the text in order to lessen the reference to human IBD in the revised manuscript (page 11).

Reviewer #5 (Remarks to the Author):

Overall this is a very nice work and I highly recommend acceptance as it is now in the revised form.

I only checked the arguments and the rebuttal of reviewer 3 (biophysical(small molecule issue). I think the authors have addressed all questions nicely and improved the clarity of the presentation.

No further changes are necessary.

We are glad for the positive assessment of our study and for your kind recommendation to the publication of this manuscript.